# Roles of two types of heparan sulfate clusters in Wnt distribution and signaling in Xenopus

Yusuke Mii [1,2,3], Takayoshi Yamamoto [1], Ritsuko Takada[2], Shuji Mizumoto[4], Makoto Matsuyama[5], Shuhei Yamada[4], Shinji Takada [2,3] & Masanori Taira[1]

Wnt proteins direct embryonic patterning, but the regulatory basis of their distribution and signal reception remain unclear. Here, we show that endogenous Wnt8 protein is distributed in a graded manner in Xenopus embryo and accumulated on the cell surface in a punctate manner in association with "N-sulfo-rich heparan sulfate (HS)," not with "N-acetyl-rich HS". These two types of HS are differentially clustered by attaching to different glypicans as core proteins. N-sulfo-rich HS is frequently internalized and associated with the signaling vesicle, known as the Frizzled/Wnt/LRP6 signalosome, in the presence of Wnt8. Conversely, N-acetyl-rich HS is rarely internalized and accumulates Frzb, a secreted Wnt antagonist. Upon interaction with Frzb, Wnt8 associates with N-acetyl-rich HS, suggesting that N-acetyl-rich HS supports Frzb-mediated antagonism by sequestering Wnt8 from N-sulfo-rich HS. Thus, these two types of HS clusters may constitute a cellular platform for the distribution and signaling of Wnt8.

[1] Department of Biological Sciences, Graduate School of Science, The University of Tokyo, 7-3-1 Hongo, Bunkyo-ku, Tokyo 113-0033, Japan. [2] National Institute for Basic Biology and Okazaki Institute for Integrative Bioscience, National Institutes of Natural Sciences, 5-1 Higashiyama, Myodaiji-cho, Okazaki, Aichi 444-8787, Japan. [3] Department of Basic Biology, School of Life Science, The Graduate University for Advanced Studies (SOKENDAI), Okazaki, Aichi 444-8787, Japan. [4] Department of Pathobiochemistry, Faculty of Pharmacy, Meijo University, 150 Yagotoyama, Tempaku-ku, Nagoya, Aichi 468-8503, Japan. [5] Division of Molecular Genetics, Shigei Medical Research Institute, 2117 Yamada, Minami-ku, Okayama 701-0202, Japan. Yusuke Mii and Takayoshi Yamamoto contributed equally to this work. Correspondence and requests for materials should be addressed to S.T. (email: stakada@nibb.ac.jp) or to M.T. (email: m_taira@bs.s.u-tokyo.ac.jp)

The Wnt family of secreted proteins plays pivotal roles in metazoan development, stem cell maintenance, and carcinogenesis[1–4]. Wnt has long been considered as a morphogen, which is a diffusible substance that gives positional information to cells and determines the spatial patterning of the embryo in a concentration-dependent manner[5–7]. Although several models for transmission/distribution of Wnt proteins have been proposed[6, 7], the molecular basis of how Wnt interacts with the cell surface for its distribution and signaling remains elusive.

Wingless (Wg, the ortholog of vertebrate Wnt1) protein forms a long-range gradient in dorsoventral patterning of the wing disc of *Drosophila*[8]. However, it should be carefully considered whether Wnt proteins act as long-range morphogens in vertebrates as well as *Drosophila*[9]. In the *Xenopus* embryo, our previous study showed that exogenous Wnt8 and Wnt11 proteins are intrinsically distributed locally[10]. Also, in vivo distribution of Wnt3 shows a short-range gradient in the intestinal stem-cell niche in mice[11]. Live-imaging of exogenous fluorescently tagged Wnt proteins in zebrafish reveals that "Wnt puncta" move along filopodial protrusions extended from the source cells and become locally located on the plasma membrane of the receiving cells[12, 13]. These results imply that the regulation of Wnt distribution appears to be intricate and context dependent or ligand dependent.

Concerning the regulatory basis for Wnt distribution, several studies have been reported. For instance, carrier molecules can modulate Wnt distribution by forming protein complexes with secreted Wnt-interacting proteins, such as secreted Frizzled-related protein (sFRP), Frzb[10, 14, 15] and a lipocalin, Swim[16], and by being loaded onto lipoprotein particles[17] or membranous carrier vehicles such as exosomes[18]. Cell-surface molecules, e.g., heparan sulfate (HS) proteoglycans (HSPGs), have also been considered to play pivotal roles in the distribution and signaling of Wnt[6]. HSPGs consist of a core protein and several HS glycosaminoglycan (GAG) chains[19, 20]. Among HSPGs, the core protein glypican and HS chains have been demonstrated to be required for the distribution and signaling of Wg in *Drosophila*, as demonstrated by the results of genetic analyses of glypicans (Dally and Dally-like)[6, 21, 22], the co-polymerase Ext[23], and the modification enzyme *N*-deacetylase/*N*-sulfotransferase (NDST)[6, 24]. In *Ndst1*-deficient mice, reduction of Wnt signaling has also been reported[25].

HS is synthesized as a co-polymer of a disaccharide unit consisting of glucuronic acid and *N*-acetylglucosamine (GlcA–GlcNAc); and this nascent precursor chain is modified through a stepwise process. The initial modification is *N*-sulfonation of GlcNAc (also commonly referred to as N-sulfation), which is catalyzed by NDSTs, followed by the epimerization of GlcA to iduronic acid (IdoA) and *O*-sulfations[6, 19, 20]. Based on biochemical analyses, it has been proposed that a single HS chain consists of two types of domain, NA and NS, which correspond to *N*-acetyl [GlcA–GlcNAc]$_n$ (unmodified domain) and *N*-sulfo [GlcA/IdoA-GlcNS]$_n$ (modified domain), respectively ("NA/NS domain model", Fig. 1a, b)[19, 20, 26]. Despite intensive genetic and biochemical studies, how HSPGs work as the interface for the distribution and signaling of Wnt proteins on the cell surface still remains vague due to the lack of cytological studies, especially for HS on the cell membrane.

To understand the spatial distribution of Wnt proteins, we examined the in vivo distribution of Wnt protein and HS chains by utilizing antibodies specific for *Xenopus* Wnt8 (generated in this study) and two types of anti-HS antibodies, NAH46 for *N*-acetyl HS[27] and HepSS-1 for *N*-sulfo HS[28]. In this study, we show that endogenous Wnt8 was distributed in a punctate manner in the extracellular space, and that, unexpectedly, "*N*-acetyl-rich HS" and "*N*-sulfo-rich HS" formed distinct clusters

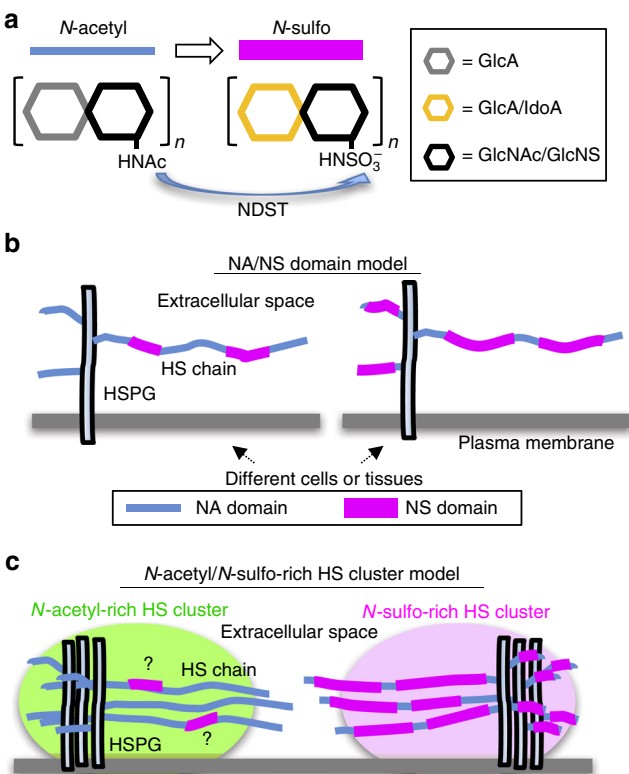

**Fig. 1** Models for heparan sulfate chain structures/organizations. **a** NDST (*N*-deacetylase/*N*-sulfotransferase). The first modification step of the heparan sulfate (HS) precursor chain is the conversion of *N*-acetyl moiety (NA) to *N*-sulfo moiety (NS), which is catalyzed by NDST. **b** The NA/NS domain model of a single HS chain. A single chain of HS generally contains two types of domains, *N*-acetyl (NA) and *N*-sulfo (NS). **c** Our model of distinct *N*-acetyl-rich HS and *N*-sulfo-rich HS clusters. See the main text for details.

on the cell membrane, named *N*-acetyl-rich HS and *N*-sulfo-rich HS clusters, respectively (Fig. 1c). Importantly, the distribution of Wnt8 puncta highly correlated with *N*-sulfo-rich HS clusters, but not with *N*-acetyl-rich HS clusters; and *N*-sulfo-rich HS clusters are incorporated into the Frizzled/Wnt/LRP6 signalosome, which are endosomes transducing Wnt signaling[29]. These findings prompted us to revisit the role of HS in the distribution and signaling of Wnt. Further analysis uncovered a question of how Wnt distribution and signaling were regulated by Frzb, a Wnt-interacting protein[10, 15], by finding that Frzb is associated with *N*-acetyl-rich HS clusters and recruits Wnt8 to *N*-acetyl-rich HS clusters.

## Results

**Endogenous Wnt8 is distributed as puncta.** The distribution of endogenous Wnt in vertebrates has rarely been examined, mainly because of the lack of antibodies available for immunostaining. However, we successfully observed endogenous Wnt8 in *Xenopus* embryos by immunohistochemistry using polyclonal antibodies generated by us (Supplementary Fig. 1 for antibody specificity). At the gastrula stage, while *wnt8* is expressed in the ventral to lateral mesoderm (Fig. 2a, left)[30], the Wnt8 protein was detected not only in the overlying ectoderm of ventral and lateral marginal zones (VMZ and LMZ, respectively) but also in the adjacent dorsal marginal zone (DMZ), forming a gradient from the LMZ toward the midline (Fig. 2a, right and Fig. 2b; see Fig. 2c, f, i and

Supplementary Fig. 11g for the VMZ). Precise observations at a higher resolution uncovered that Wnt8 did not show a continuous distribution but formed puncta at the cell boundary and inside the cells (Fig. 2c, left, indicated by white and orange arrowheads, respectively). This punctate staining was reduced by the injection of antisense morpholino oligos (MO) targeting *wnt8* (Supplementary Fig. 1c) and was not observed with pre-immune serum (Fig. 2c, right), showing that the staining was specific for Wnt8. Such punctate distribution was also detected in live-imaging of Wnt8 fused with monomeric Venus (mV) (mV-

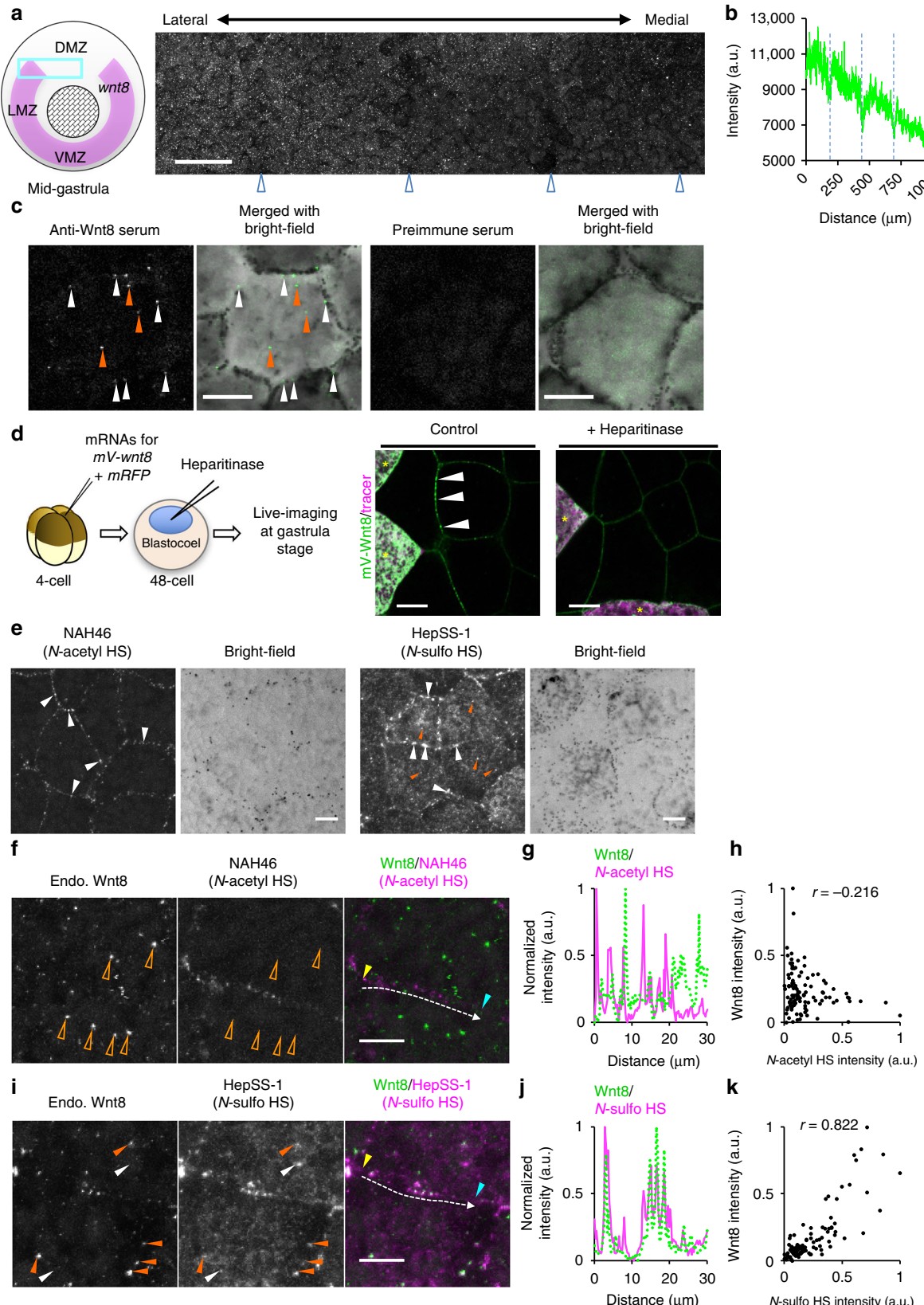

Wnt8) (Fig. 2d, arrowheads in control; Supplementary Fig. 2a for biological activity). Furthermore, immunostaining of Myc-tagged Wnt8 (Wnt8-Myc) with ZO-1 showed that Wnt8 puncta were localized on the basolateral side of the intercellular space below the tight junction indicated by ZO-1 staining[31] (Supplementary Fig. 2b).

**Wnt8 is colocalized with a subpopulation of HS chains.** The punctate distribution of Wnt8 prompted us to search for docking molecules on the cell surface for Wnt8 accumulation. One of the candidates was the HS chain, because HS or HSPGs are shown to be essential for the proper distribution of secreted signaling proteins, including Wnt[6, 19]. To test this possibility we removed HS chains from the cell surface by blastocoel injection of a mixture of heparitinase I and heparitinase II, which cleaves a wide range of HS structures[32]. The heparitinase treatment greatly reduced punctate signals of mV-Wnt8 on the cell boundary (Fig. 2d, right), suggesting that HS is critical for the punctate distribution of Wnt8 in the intercellular space.

To examine the organization of HS chains on the cell surface in *Xenopus* gastrula embryos, we visualized HS chains by using two types of anti-HS antibodies, NAH46, which recognizes *N*-acetyl HS (several consecutive *N*-acetylated disaccharide units)[27] and HepSS-1, which recognizes *N*-sulfo HS (several consecutive *N*-sulfated disaccharide units with little or no *O*-sulfation)[28, 33]. Strikingly, both antibodies stained punctate structures or "clusters" at the cell boundary, but the staining patterns were quite different from each other (Fig. 2e, arrowheads, see also Supplementary Fig. 3a–c). Of note, co-immunostaining of embryos with anti-Wnt8 and these anti-HS antibodies showed that the punctate staining of endogenous Wnt8 was highly colocalized with that of HepSS-1 but not NAH46 (Fig. 2f–k, see also Supplementary Fig. 2c for low magnification images). In addition, the colocalization of Wnt8 and HepSS-1 staining was also observed inside the cells (Fig. 2i, Supplementary Fig. 2d, orange arrowheads). The correlation coefficients ($r$) of the distributions of Wnt8 and *N*-sulfo HS ($0.784 \pm 0.044$ (mean $\pm$ s.d.); $n = 6$) were significantly higher ($p < 0.001$, *t*-test) than those of Wnt8 and *N*-acetyl HS ($0.028 \pm 0.097$ (mean $\pm$ s.d.), $n = 6$). These data suggest that a certain subset of HS chains play a role in docking Wnt8 on the cell surface and internalizing Wnt8 inside the cell.

**N-acetyl-rich and N-sulfo-rich HS form distinct clusters.** To better understand the clusters of HS chains stained with NAH46 and HepSS-1, we carefully examined the distribution of these HS chains by confocal analysis at the subapical region of cells (basal to the tight junction) in the superficial ectodermal layer. Firstly, by double staining using directly fluorescence-labeled NAH46

and HepSS-1 antibodies we confirmed that the punctate stainings of the two antibodies were differentially distributed on the cell boundary in a single cell (Fig. 3a). Quantification of fluorescent intensities along the cell boundary and scatter plot analysis revealed that the correlation coefficient ($r$) of their distributions is almost zero (Fig. 3b, c; $r = 0.031$), indicating that the clusters stained with NAH46 and HepSS-1 are distinct. The difference of the staining patterns was also observed in a lower magnification; NAH46 staining was almost consistent among cells, whereas HepSS-1 staining varied among cells (Supplementary Fig. 3a, b). The clusters stained with both antibodies at the cell boundary were likely to be present in the extracellular space, because heparitinase introduced by blastocoel injection greatly reduced the staining (Supplementary Fig. 3c). Secondly, we found that the intracellular staining was specific for HepSS-1. In contrast, NAH46 staining was largely limited to the cell boundary (Fig. 2e). Because HepSS-1 staining inside the cell was reduced by treatment with heparitinase, which is not cell-permeable (Supplementary Fig. 3c, orange arrowheads), the intracellular HepSS1-positive puncta are likely to be derived from the cell membrane. This issue will be addressed later.

We also observed that low levels of NAH46 signals (*N*-acetyl HS) were widely distributed along the cell boundary (Fig. 3b). This implies that not all NAH46 signals form into clusters. This also suggests that HepSS-1-positive HS chains contain NAH46 epitopes to some extent, being consistent with the NA/NS domain model (Fig. 1b). The size of both types of HS clusters were estimated to be around 200 nm in diameter by stimulated emission depletion (STED) microscopy[34, 35] (Supplementary Fig. 3d, e). Distinct distributions of NAH46 and HepSS-1 staining puncta were also observed in the HeLa cells on the cell surface (Supplementary Fig. 3f, g), indicating the generality of the organization of HS clusters (Fig. 1c).

To examine whether the HS clusters recognized with NAH46 and HepSS-1 are actually enriched with *N*-acetyl and *N*-sulfo HS, respectively, we examined the effect of *N*-deacetylase/*N*-sulfotransferase 1 (Ndst1, see Fig. 1a) on the staining with these two antibodies. It is expected that gain- or loss-of function of *ndst1* can increase or decrease *N*-sulfo HS (i.e., decrease or increase *N*-acetyl HS), respectively, in *Xenopus* embryos (for *ndst1* expression patterns, see Supplementary Fig. 4a). We injected *ndst1* mRNA into one blastomere at the 4-cell stage and observed the two types of HS clusters at the gastrula stage in the region where *ndst1* overexpressing and intact cells existed mosaically. Overexpression of *ndst1* significantly increased *N*-sulfo HS and decreased *N*-acetyl HS staining (Fig. 3d; see Supplementary Fig. 4b for the quantification method). Conversely, knockdown of *ndst1* using MO significantly decreased *N*-sulfo, and increased *N*-acetyl HS staining (Fig. 3e). These effects of *ndst1* MO were rescued by

**Fig. 2** Endogenous Wnt8 shows not only a gradient but also HS-dependent punctate distributions. **a** The gradient of the endogenous Wnt8 protein from the lateral to mid-dorsal marginal zones at the mid-gastrula stage (st. 11.5). The observed region is indicated by the cyan box (reported localization of *wnt8* mRNA in magenta). Embryos were flat-mounted under a coverslip and the image was acquired using automatic tiling with the maximum intensity projection of *z*-stacks (junctions of tiling appeared darker, arrowheads). **b** Quantification of the gradient of Wnt8 staining (**a**). The junctions of tiling are indicated by dotted lines, showing the drop in intensity. **c** A high-resolution image of endogenous Wnt8 staining in the VMZ. *wnt8* is expressed in the underlying mesoderm, but not in the superficial layer in *Xenopus* embryos. An optical section at the subapical region of cells (basal to the tight junction) of the superficial layer is shown (see also Supplementary Fig. 2b, d). **d** Exogenous mV-Wnt8 expression and heparitinase treatment. Experimental procedures are illustrated on the left. Embryos were observed at stage 10.5. The source cells are indicated with "*". **e** Immunostaining of HS chains with NAH46 (for *N*-acetyl HS) or HepSS-1 (for *N*-sulfo HS). Notably HepSS-1 staining shows puncta inside cells (orange arrowheads). **f–k** The colocalization of endogenous Wnt8 and a subpopulation of HS. Gastrula embryos (st. 10.5) were co-immunostained for Wnt8 and NAH46 epitope (*N*-acetyl HS) or HepSS-1 epitope (*N*-sulfo HS) at the VMZ (**f**, **i**). Signal intensities along white arrows were plotted (**g**, **j**), starting and ending points as indicated by yellow and cyan arrowheads, respectively (**f**, **i**). Scatter plots show the indicated signal intensities for every pixel along the arrow, presented with correlation coefficients ($r$) (**h**, **k**). Note *N*-sulfo-rich HS clusters inside cells, with Wnt8 (orange arrowheads) and without Wnt8 (white arrowheads). Staining of *N*-acetyl HS was absent from the Wnt8 puncta inside cells (open arrowheads in **f**). Images are a representative of at least two independent experiments. Amounts of mRNA (ng/embryo): *mV-wnt8*, 1.0; *mRFP*, 0.50. Scale bars, 100 μm (**a**); 10 μm (**c–f**, **i**). a.u., arbitrary units

addition of a small amount of *ndst1* mRNA (Fig. 3e). Thus, *ndst1* is necessary and sufficient for *N*-sulfo HS formation in *Xenopus* embryos. Based on the cytological data shown above, we refer to the punctate structures immunostained with NAH46 and HepSS-1 as "*N*-acetyl-rich HS clusters" (visualized by NAH46) and "*N*-sulfo-rich HS clusters" (visualized by HepSS-1) (Fig. 1c).

**N-acetyl- and N-sulfo-rich HS are biochemically separable.** To examine biochemically the subtypes of HS chains, we took two approaches, immunoprecipitation followed by disaccharide analysis and cellulose acetate membrane electrophoresis (CAME). We undertook CAME, which has been used for analysis of GAGs[36] (Supplementary Fig. 5a), and combined it with a newly developed immunodetection system.

Since cellulose acetate membrane does not have molecular sieving propaties, GAGs are expected to migrate according to their net charge density but not their chain length (molecular weight) in CAME analysis. We first monitored migration of standard GAGs in the CAME. As standard GAGs, we used heparosan[39] from *Escherichia coli* ([GlcA–GlcNAc]$_n$, unsulfated chain), completely de-sulfated, *N*-acetylated (CDSNAc) HS, and completely de-

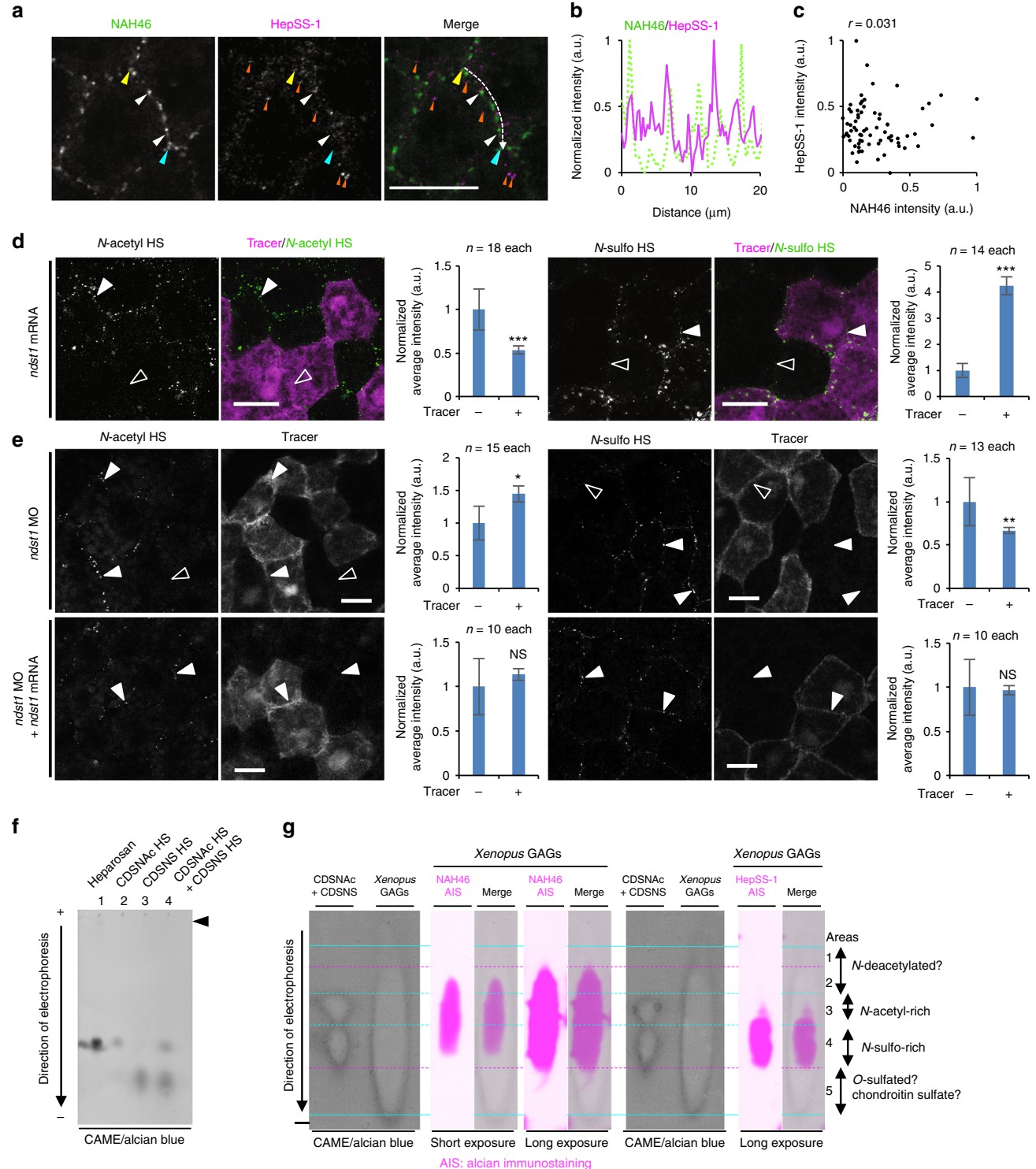

sulfated, N-sulfated (CDSNS) HS (Supplementary Fig. 5b). As expected, heparosan and CDSNAc HS, both of which completely lack the sulfation, showed similar mobility, whereas CDSNS HS, which is highly N-sulfated, showed higher mobility than heparosan and CDSNAc HS, indicating that GAGs were separated according to their charge density (Fig. 3f). The data also confirmed that the chain length does not affect the migration of GAGs in this system because heparosan sample was migrated as a single spot in spite of the heterogeneity in its chain length (Fig. 3f)[37].

Using this system, we analyzed GAGs purified from Xenopus gastrula embryos (st. 11.5). As shown in Fig. 3g (left panel), Xenopus GAGs were widely separated as visualized by alcian blue staining. For detection of specific HS chains after CAME, we combined it with a new immunodetection method, named "CAME coupled with alcian immunostaining (CAME-AIS)", which is similar to western blot analysis. In this method, weak staining with alcian blue serves as both visualization and fixation of electrophoresed GAGs (Supplementary Fig. 5c, d). In AIS, HepSS-1 signal was mostly detected in Area 4, which roughly corresponded to the area of the standard CDSNS HS, and only weakly in a relatively upper area (Area 3) (Fig. 3g). In contrast, strong NAH46 signal covered in Area 3 (Fig. 3g, short exposure), which was surrounding the area of the standard CDSNAc HS, and weak but evident signal was also detected in Areas 2 and 4 as well (Fig. 3g, long exposure). Thus, HS chains in Area 3 (CDSNAc HS equivalent) and Area 4 (CDSNS HS equivalent) were likely to correspond to N-acetyl-rich HS and N-sulfo-rich HS, respectively, according to their mobilities (negative charge density) and the distributions of NAH46 and HepSS-1 signals. The different migration patterns of NAH46 and HepSS-1 positive HS chains in CAME-AIS analysis appeared to well correspond to our cytological data showing distinct clustering of N-acetyl-rich and N-sulfo-rich HS chains (Fig. 3a–c). On the other hand, wide distribution of NAH46-positive HS suggests that the content of modifications in a single GAG chain is variable. Especially we note that even N-sulfo-rich HS, detected in Area 4, appears to contain some consecutive N-acetyl disaccharide units. This result is also consistent with our cytological observation showing wide distribution of low levels of NAH46 staining along the cell boundary in Xenopus embryos (Fig. 3b).

Taken together, the cytological and biochemical analysis reveals three major findings on the organization of HS chains as follows. The heterogeneity of HS chains exists in a single cell, and the two types of HS chains are not uniformly distributed, but form clusters on the cell surface. Furthermore, N-acetyl-rich and N-sulfo-rich HS clusters are discretely present on the cell surface (Fig. 1c).

**Glypicans as core proteins for N-acetyl/N-sulfo-rich HS.** To identify core proteins for the two types of HS clusters, we focused on cell surface HSPGs, which are categorized into two types, transmembrane proteins (syndecans, etc.) and the GPI-anchored protein glypicans[6]. To discriminate these two types of HSPGs, we used phosphatidylinositol phospholipase C (PI-PLC), which cleaves GPI-anchored proteins[38, 39]. Extracellular application of the PI-PLC enzyme by blastocoel injection reduced the staining of both N-sulfo-rich and N-acetyl-rich HS clusters (Fig. 4a). To compare cells side-by-side, we constructed a Golgi-tethered form of PI-PLC (gPI-PLC) to express PI-PLC in a cell-autonomous manner, and found that the staining of both N-sulfo-rich and N-acetyl-rich HS clusters was clearly reduced at the boundary between gPI-PLC expressing cells (Supplementary Fig. 6a, open arrowheads). Because GPI-anchored proteins localize to lipid rafts[40], we applied methyl-β-cyclodextrin (MβCD) to Xenopus embryos to disrupt lipid rafts by removing cholesterol[41]. The data showed that MβCD treatment reduced both N-sulfo-rich and N-acetyl-rich HS clusters in a dose-dependent manner (Supplementary Fig. 6b). Thus, glypicans are likely to be core proteins of the two types of HS clusters.

We focused on glypican 4 (gpc4) and gpc5, because they are highly expressed in the early Xenopus embryo (Supplementary Fig. 6c)[42, 43]. The single MO-mediated knockdown of gpc4, but not gpc5, decreased N-acetyl-rich HS clusters, whereas the double knockdown of gpc4 and gpc5, but neither alone, decreased N-sulfo-rich HS clusters (Fig. 4b, see also Supplementary Fig. 7a; Supplementary Fig. 6d for the specificity of MOs). Consistently, overexpression of either gpc4 or gpc5 increased N-sulfo-rich HS clusters (Fig. 4c). Of note, overexpression of gpc4, but not gpc5, increased N-acetyl-rich HS clusters (Fig. 4c). As negative controls, overexpression of ΔHS mutants of gpc4 or gpc5, whose products lack putative HS-attachment sites and supposedly do not have HS chains, did not affect both types of HS clusters (Supplementary Fig. 7b). These data suggest that Gpc4 is a core protein of both N-sulfo-rich and N-acetyl-rich HS clusters, whereas Gpc5 is only a core protein of N-sulfo-rich HS clusters.

**N-sulfonation-dependent internalization of the HS clusters.** We have already inferred that N-sulfo-rich clusters appear to be frequently internalized into the cell compared to N-acetyl-rich HS clusters (see Supplementary Fig. 3c). To directly compare the internalization of N-sulfo-rich HS clusters with that of N-acetyl-rich, we injected NAH46 or HepSS-1 antibodies into the blastocoel of early blastula, and observed the animal cap region, where Wnt signaling is considered to be low[11], at the gastrula stage after antibody staining. The data showed that NAH46 was detected on the cell boundary (Fig. 5a, left), similar to standard immunostaining with NAH46 (Fig. 2e), whereas HepSS-1 was hardly detected on the cell boundary (Fig. 5a, right), suggesting that N-sulfo-rich HS clusters are frequently removed from the cell

**Fig. 3** The two types of HS clusters and CAME-AIS analysis of Xenopus GAGs. **a–c** The distinct distributions of the two types of HS at a cell boundary in the DMZ. Xenopus gastrula (st. 11.5) embryos were immunostained with directly fluorescent-labeled NAH46 and HepSS-1 antibodies. White arrowheads, overlap of NAH46 and HepSS-1 staining. Orange arrowheads, HepSS-1 staining inside the cell. Fluorescence intensities of N-acetyl HS (NAH46 epitope, green) and N-sulfo HS (HepSS-1 epitope, magenta) were quantified similarly to Fig. 2f–k (**b**, **c**). **d**, **e** ndst1 mRNA (**d**) or MO (**e**) was coinjected with mRFP mRNA or FITC-dextran, respectively, as a tracer into the animal pole region of a ventral blastomere at the 4- or 8-cell stage. Injected embryos were fixed at st. 11.5, and immunostained for N-acetyl-rich or N-sulfo-rich HS. The staining intensities of HS were measured as exemplified in Supplementary Fig. 4b and normalized average intensities of the HS staining are presented as graphs (mean±s.e.m.). Statistical analysis was performed with t-test (*p < 0.05, **p < 0.01, ***p < 0.001, NS, not significant). Numbers of measured cell boundaries were as indicated on the top of the graphs. **d** Effects of ndst1 overexpression on the HS clusters. White arrowheads indicate stronger staining compared with open arrowheads. **e** Effects of ndst1 knockdown with ndst1 MO on the HS clusters. Upper panels, white arrowheads indicate stronger staining compared with open arrowheads. Lower panels, white arrowheads indicate no apparent difference in the HS staining at the boundary between tracer-negative or -positive cells. **f** Cellulose acetate membrane electrophoresis (CAME) of GAGs. Indicated GAGs were stained with alcian blue. **g** CAME-alcian-immunostaining (CAME-AIS) of Xenopus GAGs. Based on the charge-densitydependent separation of CDSNAc HS and CDSNS HS, Areas 1 through 5 were determined. It is likely that HS in Area 3 corresponds to N-acetyl-rich HS, and that HS in Area 4 corresponds to N-sulfo-rich HS. Images are a representative of at least two independent experiments. Amounts of mRNA (ng/embryo): mRFP, 0.5; ndst1, 0.020 (**d**), or 0.013 (**e**). Amounts of MOs (ng/embryo): std MO and ndst1 MO, 14. Scale bars, 20 μm. a.u., arbitrary units

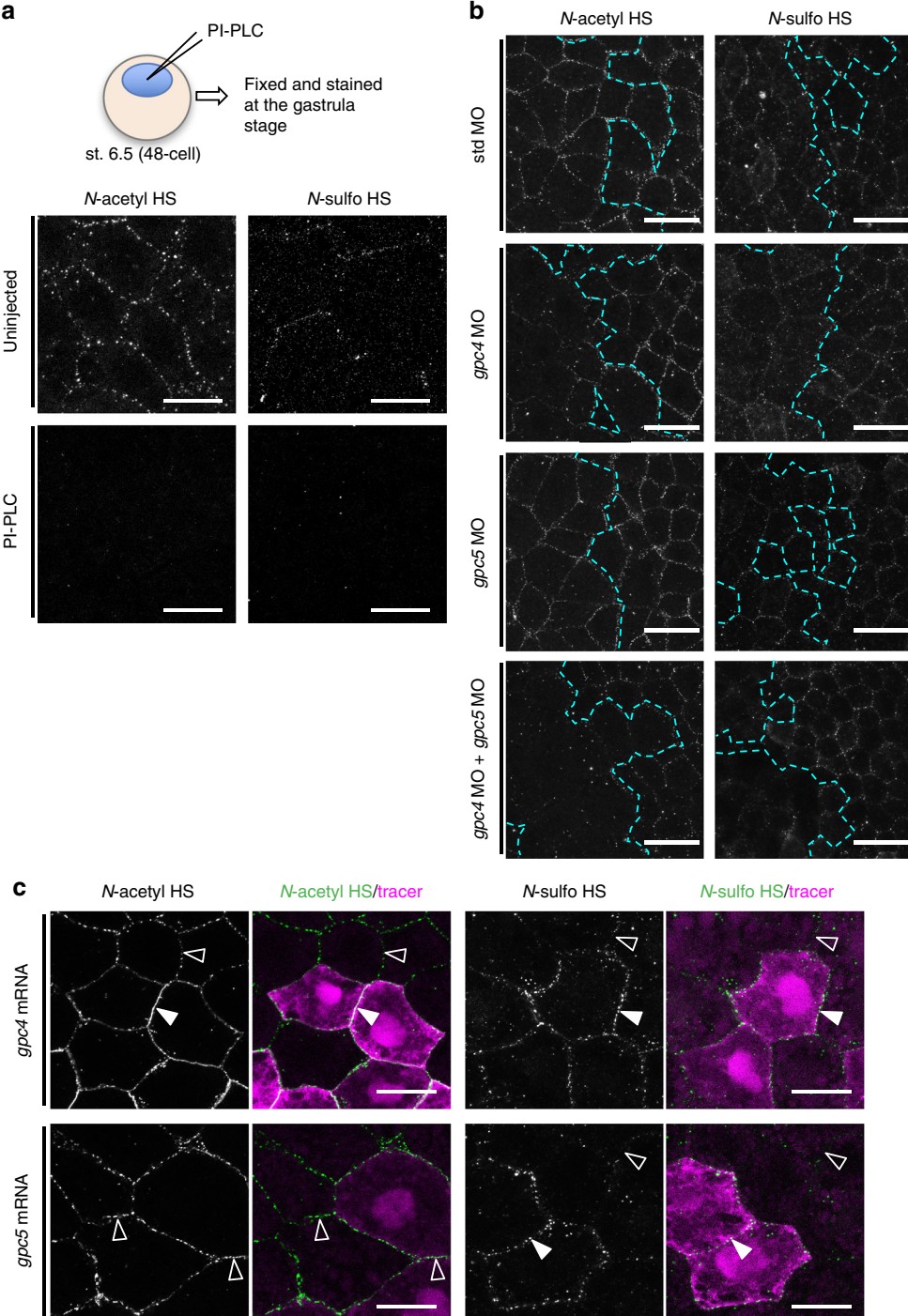

**Fig. 4** Glypicans are core proteins for *N*-acetyl/*N*-sulfo-rich HS clusters. **a** The reduction in the numbers of both *N*-sulfo-rich and *N*-acetyl-rich HS clusters by PI-PLC treatment. Embryos were fixed and stained at st. 11.5 after blastocoel-injection of PI-PLC (800 μU) at st. 6.5. **b** Effects of the knockdown of *gpc4*, *gpc5* or both on the formation of HS clusters. MOs with FITC-dextran were injected into a dorsal blastomere of 4-cell stage embryos. Boundaries of the tracer-positive regions (left side) are indicated by dotted cyan lines (see Supplementary Fig. 7a for tracer images). The reduction of *N*-acetyl-rich HS clusters was observed with *gpc4* MO alone, whereas that of *N*-sulfo-rich HS clusters was apparent only with the combination of *gpc4* MO and *gpc5* MO. Amounts of MOs (ng/embryo), std MO, 28; *gpc4* MO and *gpc5* MO, 14. **c** Effects of the overexpression of *gpc4* or *gpc5* on the formation of HS clusters. mRNA for *gpc4* or *gpc5* was coinjected with *mRFP* mRNA into the animal pole region of a ventral blastomere at the 4- or 8-cell stage and fixed at the gastrula stage (st. 11.5) and stained. Upper panels, exogenous *gpc4* expression increased the number of both *N*-sulfo-rich and *N*-acetyl-rich HS clusters (white arrowheads), compared with the cell boundaries of tracer-negative cells (open arrowheads). Lower panels, *gpc5* expression did not increase the number of *N*-acetyl-rich HS clusters (open arrowheads) but did increase the number of *N*-sulfo-rich HS clusters (white arrowhead). Amounts of injected mRNAs (pg/embryo): *gpc4* and *gpc5*, 50; *mRFP*, 400. See also Supplementary Fig. 7b for *gpcΔHS* mutants. Images are a representative of at least two independent experiments. Scale bars, 20 μm (**a**, **c**); 40 μm (**b**)

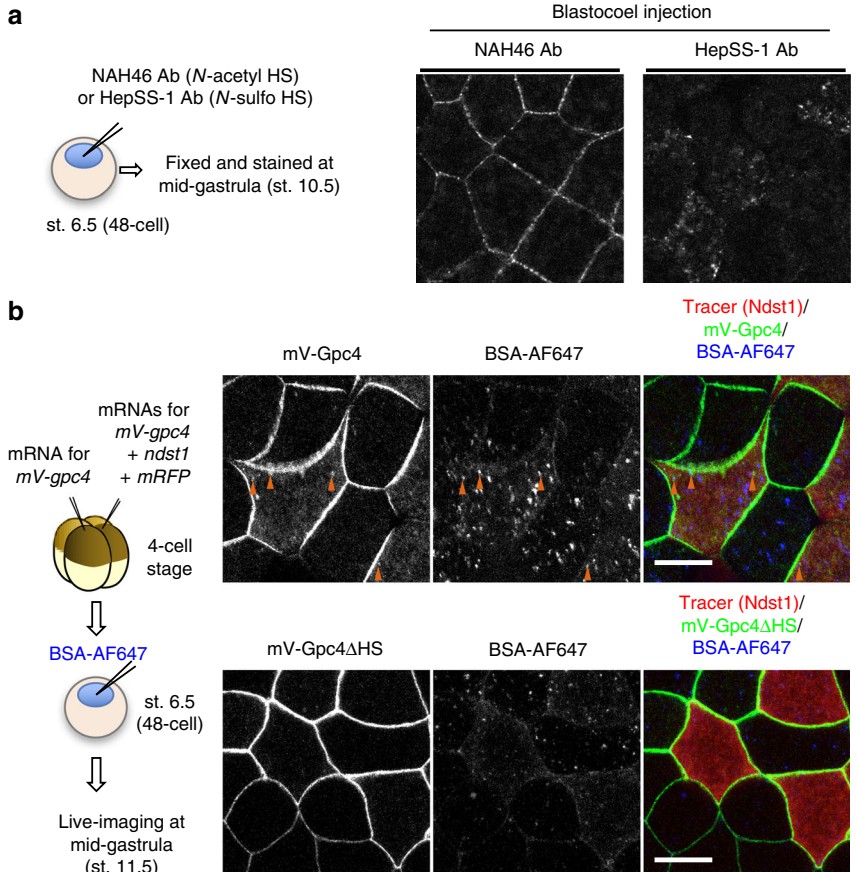

**Fig. 5** *N*-sulfonation-dependent internalization of the HS clusters. **a** Internalization assay of anti-HS antibodies. NAH46 or HepSS-1 antibody (10 ng) was injected into blastocoels at st. 6.5, and the injected embryos were fixed at st. 10.5 and stained with anti-mouse IgM secondary antibody. The animal cap region, where Wnt signaling is considered to be low[10], was observed. Distribution of the injected anti-*N*-acetyl HS antibody showed an intercellular distribution similar to normal immunostaining of gastrula with the anti-*N*-acetyl HS antibody (Fig. 2e, Supplementary Fig. 3c), whereas that of the anti-*N*-sulfo HS antibody showed puncta inside the cells. **b** Facilitated internalization of the mV-Gpc4 core protein by coexpression of *ndst1*. Dye-labeled BSA (BSA-AF647) was injected into the blastocoel as a marker for internalized puncta. The animal cap region was observed at st. 11.5. Upper panels, in *ndst1*-overexpressing cells, the number of puncta of mV-Gpc4 that overlapped with BSA-AF647 (orange arrowheads) increased. Lower panels, Ndst1 did not facilitate the internalization of mV-Gpc4ΔHS, which lacks HS chains. Amounts of injected mRNA (ng/embryo): *mV-gpc4*, 0.067 × 2; *ndst1*, 0.033; *mRFP*, 0.33. Images are a representative of at least two independent experiments. Scale bars, 20 μm. a.u., arbitrary units

surface by endocytosis and degradation. This internalization required the attachment of sugar chains because Gpc4ΔHS showed no punctate staining inside any cells (Fig. 5b, lower panels). Furthermore, this internalization was mediated by *N*-sulfonation of HS chains, because *ndst1* overexpression increased mV-Gpc4 in some endosome populations that could be visualized by dye-labeled bovine serum albumin (BSA-AF647) which had been injected into the blastocoel (Fig. 5b, upper panels). These results suggest that the HS clusters are internalized from the cell surface in an *N*-sulfonation-specific manner.

**Ndst1 is involved in distribution and signaling of Wnt8**. Since Wnt8 is highly colocalized with *N*-sulfo-rich HS clusters (Fig. 2f–k), we next addressed whether the N-sulfonation of HS is involved in Wnt signaling. Firstly, by overexpression or knockdown of *ndst1*, we examined *N*-sulfo-rich HS as a docking site for Wnt8. The data showed that *ndst1*-overexpressing cells accumulated mV-Wnt8 or endogenous Wnt8 on the cell boundary (Fig. 6a, b), whereas *ndst1* MO decrease it (Fig. 6c). Similar results were also obtained in HeLa cells with Wnt3a (Supplementary Fig. 12a–d). These data demonstrate that Wnt8 and Wnt3a, both of which mainly activate canonical Wnt signaling, preferentially

accumulate on *N*-sulfo-rich, but not on *N*-acetyl-rich HS clusters in *Xenopus* embryos and HeLa cells.

As predicted from our conclusion of Gpc4 and Gpc5 as core proteins for *N*-sulfo-rich HS (Fig. 4), both *gpc4*- and *gpc5*-overexpressing cells accumulated mV-Wnt8 on the cell membrane (Supplementary Fig. 8a). It was reported that Wnt8 directly binds to glypican core proteins[44], but cells expressing the ΔHS mutants of *gpc4* and *gpc5* did not accumulate mV-Wnt8 (Supplementary Fig. 8a), further supporting our conclusion that *N*-sulfo-rich HS clusters accumulates Wnt8 through HS chains, not core proteins.

We next examined the role of *N*-sulfo-rich HS clusters in activation of canonical Wnt signal transduction. In *ndst1*-overexpressing cells, nuclear localization of β-catenin was significantly upregulated (Fig. 6d–f), whereas *ndst1* MO caused a statistically significant downregulation of nuclear localization of β-catenin (Fig. 6d, g, h). Similarly, TOP-Flash reporter analysis[45] showed that a high level of *ndst1* expression (500 pg/embryo), but not low levels (20 or 50 pg/embryo), significantly upregulated the TOP-Flash reporter activity (Fig. 6i, j). Conversely, *ndst1* MO significantly reduced the reporter activity, which can be rescued with addition of the lower amounts of *ndst1* mRNA (20 or 50 pg/embryo) (Fig. 6k). Furthermore, the knockdown of *ndst1* inhibited secondary axis formation induced by the ventral

injection of *wnt8* mRNA (Supplementary Fig. 8b). In addition, the expression of *gbx2*, a direct Wnt target gene[46], was reduced by *ndst1* knockdown (Supplementary Fig. 8c). These data indicate that *ndst1* is critical for both *N*-sulfo-rich HS cluster formation and canonical Wnt signaling, thereby suggesting that the *N*-sulfo-rich HS cluster is crucial for Wnt signaling.

**N-sulfo-rich HS as a pre-existing core for the Wnt signalosome.** As endogenous Wnt8 was colocalized with *N*-sulfo-rich HS inside

cells (see Fig. 2i, orange arrowheads), we hypothesized that the internalization of *N*-sulfo-rich HS clusters (Fig. 5a) is involved in the signal transduction of Wnt by functioning in the formation of the Frizzled/Wnt/LRP6 signalosome[1, 2, 29, 47]. We first examined whether Wnt8 is internalized in association with *N*-sulfo-rich HS clusters . To this end, tagged Wnt8 (mV-Wnt8 or Wnt8-Myc) was expressed to distinguish Wnt-expressing cells and Wnt-receiving cells. In this system, Wnt puncta inside the receiving cells must be internalized from the outside because we observed the subapical region of the cells (basal to the tight junction, not

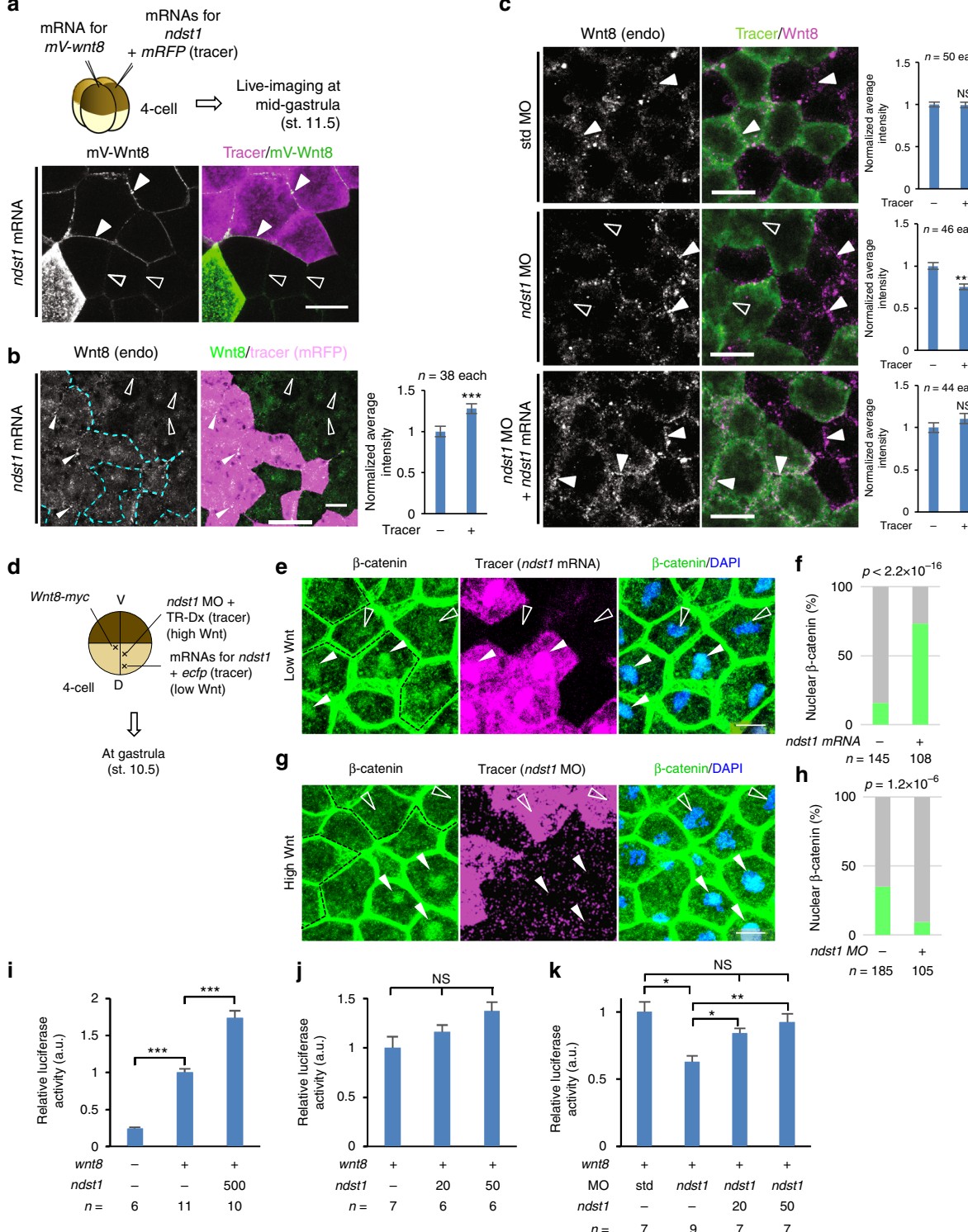

the cell surface), by confocal microscope (see Supplementary Fig. 11i for the depth of the optical sections; Supplementary Fig. 2b for the basolateral distribution of Wnt8-Myc). As shown by time-lapse confocal imaging in *Xenopus* embryos (Fig. 7a and Supplementary Movie 1), puncta of mV-Wnt8 on the cell membrane were internalized into the cell within seconds (cyan arrowheads), similar to an observation in zebrafish[48]. Internalized puncta of Wnt8-Myc mostly overlapped with *N*-sulfo-rich HS, but not with *N*-acetyl-rich HS (Fig. 7b, open arrowheads for no overlap and closed arrowheads for overlap).

The internalization of Wnt8 in association with *N*-sulfo-rich HS clusters raised the possibility that both Wnt8 and *N*-sulfo-rich HS clusters are included in the signalosome. To test this, we examined the colocalization of HS clusters with phosphorylated LRP6 (pLRP6)[2], which is an evident sign of signal transduction of the canonical Wnt pathway as well as a major component of the signalosome[29]. To enhance immunostaining of pLRP6, we overexpressed LPR6. By using this system, we could observe that pLRP6 staining is increased by Wnt8 in *Xenopus* embryo (Supplementary Fig. 9a). Furthermore, pLRP6 on the cell membrane was highly colocalized with *N*-sulfo-rich HS (Fig. 8d–f), but not with *N*-acetyl-rich HS (Fig. 8a–c). Phosphorylated LRP6 and *N*-sulfo-rich HS were also colocalized inside cells (Fig. 8d, white arrowheads) and puncta of pLRP6 inside cells were mostly colocalized with internalized puncta of mV-Wnt8 (Supplementary Fig. 9b). These results strongly suggest that *N*-sulfo-rich HS, as well as the Wnt8 ligand, are incorporated into the signalosome.

We next examined Frizzled8 (Fzd8, Supplementary Fig. 10 for antibody validation), another component of the signalosome as well as a receptor of Wnt8[49]. We found that endogenous Fzd8 is highly colocalized with *N*-sulfo-rich HS, but not with *N*-acetyl-rich HS clusters (Supplementary Fig. 11a-f). Interestingly, the amount of Fzd8 on the cell membrane appears to depend on Wnt8, because Fzd8 staining is higher in VMZ, where high level of Wnt8 is expressed, than in the mid-region of the DMZ (mDMZ), where Wnt8 expression is low. Therefore, Wnt8 might increase Fzd8 protein levels in the ventrolateral region, even though *fzd8* transcripts are enriched dorsally[50]. In fact, Fzd8 staining was increased in the mDMZ by overexpression of Wnt8 (Supplementary Fig. 11g). Considering the dorsally enriched *fzd8* transcripts[50], Wnt8 possibly stabilizes Fzd8 protein rather than enhances the expression of *fzd8*. Consistently, the overlap of *N*-sulfo-rich HS and Fzd8 was more frequently detected in the VMZ than in the mDMZ, and increased in the mDMZ by *wnt8* overexpression (Supplementary Fig. 11g, see middle panels of "overlap of *N*-sulfo-rich HS and Fzd8"). Similar to pLRP6, puncta of Fzd8 inside cells were well colocalized with internalized puncta of Wnt8-Myc (Supplementary Fig. 11h, i). In

contrast, Wnt8 did not affect the amount of *N*-sulfo-rich HS (Supplementary Fig. 11g, compare left-most panels). These data suggest a scenario as follows: at first, Wnt8 docking on an *N*-sulfo-rich HS cluster recruits and accumulates Fzd8 at the cluster, and thereby activates signaling components including LRP6, and then Wnt ligand and *N*-sulfo-rich HS are internalized with Wnt signaling components including Fzd8 and pLRP6 to form signalosomes. Consistent with this scenario, *ndst1* MO specifically reduced the level of membrane-localised Fzd8 (Fig. 8g). Furthermore, *ndst1* MO specifically reduced the level of pLRP6 (Fig. 8h, i), suggesting that *N*-sulfo-rich HS is involved in Wnt signaling through the phosphorylation of a signalosome component, LRP6. Even in a low Wnt activity region, *N*-sulfo-rich HS was frequently internalized as shown above (Fig. 5a). In line with the internalization of HS as well as Wnt signaling, caveolin, a critical component of endosome formation, which is required for the canonical pathway[51], showed higher colocalization with *N*-sulfo-rich HS than with *N*-acetyl-rich HS in the absence of exogenous Wnt ligands (Supplementary Fig. 12e, f). Thus, the internalization of *N*-sulfo-rich HS appears to occur even in conditions where Wnt signaling is not activated. In addition, *N*-sulfonation of HS enhanced internalization of mV-Gpc4 (Fig. 5b). Taken together, these data suggest that *N*-sulfo-rich HS clusters function as pre-existing cores for signalosome formation initiated by Wnt8 (Fig. 10d).

**Role of *N*-acetyl-rich HS with Frzb in regulation of Wnt.** Frzb, a member of the sFRP family, is widely distributed in the extracellular space[10], and not only binds to Wnt as an inhibitor[52–54] but also expands the distribution range of Wnt in the extracellular space, and its signaling range[10, 14]. This raised interesting questions as to whether Frzb colocalizes with *N*-acetyl or *N*-sulfo-rich HS clusters and whether Frzb affects the interaction between Wnt8 and *N*-sulfo-rich HS clusters. To visualize the Frzb protein, mVenus-tagged Frzb (mV-Frzb) (Supplementary Fig. 13a for biological activity) or HA-tagged Frzb (Frzb-HAi)[53] was used for direct observation or immunostaining, respectively. In the *Xenopus* embryo, mV-Frzb was distributed at the cell boundaries (Supplementary Fig. 13b, left). Notably, this distribution was HS chain-dependent as well as extracellular, because heparitinase injected into the blastocoel reduced it (Supplementary Fig. 13b, right). We therefore asked whether Frzb colocalizes with *N*-acetyl- or *N*-sulfo-rich HS clusters. For this purpose, Frzb-HAi was expressed at a lower level than mV-Frzb and was detected by immunostaining, thereby revealing punctate distribution of Frzb at the cell boundary (Fig. 9a; left-most panels). Importantly, we found that Frzb-HAi preferentially colocalized with *N*-acetyl-rich HS clusters ($r = 0.868$) compared to *N*-sulfo-rich HS clusters

**Fig. 6** Ndst1 is necessary and sufficient for distribution and signaling of Wnt8. **a** Effects of *ndst1* overexpression on mV-Wnt8 accumulation. **b** Effects of *ndst1* overexpression on endogenous Wnt8 in the VMZ. mRNAs of *ndst1* and *mRFP* were coinjected into the ventral region of a blastomere at the 4-cell stage. Dashed line, boundary of the tracer-positive cells. Graph, quantification of endogenous Wnt8 at cell boundaries (mean±s.e.m.). Number of the measured cell boundaries were as indicated. **c** Effects of *ndst1* knockdown on endogenous Wnt8 in the VMZ. MO and FITC-dextran were coinjected into the ventral region of a blastomere of 4-cell stage embryos. Graphs, quantification of endogenous Wnt8 at cell boundaries (mean±s.e.m.). The number of the measured cell boundaries (n) was as indicated. **d** Schema for overexpression and knockdown of *ndst1*. A view from the animal side. *wnt8-myc* mRNA was injected into the animal pole region. *ndst1* mRNA or *ndst1* MO was injected as indicated. TR-Dx, Texas red-conjugated dextran. **e–h** Overexpression or knockdown of *ndst1*. The number of cells with or without nuclear β-catenin staining was counted in a low Wnt activity area for overexpression (**e**, **f**) or in a high Wnt activity area for knockdown (**g**, **h**). The Nnumber of the analyzed cells was as indicated (**f**, **h**). **i–k** TOP-Flash reporter assay. Graphs, mean±s.e.m. The number of the measured pools (each pool contained three embryos) was as indicated. *Xenopus* embryos at stage 11.5 were analyzed. Images are a representative of at least two independent experiments. White arrowheads indicate increase or stronger signal compared to open arrowheads (the absence of open arrowheads indicates no difference). Statistical tests (all two-sided): *t*-test (**b**, **c**); Fisher's exact test (**f**, **h**); or pairwise Wilcoxon rank sum test (**i–k**, multiple comparison). Significance levels: *$p < 0.05$, **$p < 0.01$, ***$p < 0.001$, NS, not significant. Amounts of mRNA (pg/embryo): *mV-wnt8*, 1000; *mRFP*, 500; *ndst1*, 500 (**a**, **b**, **e**, **i**), 20 (**j**, **k**), or 50 (**j**, **k**) or 3.3 (**c**); *wnt8-myc*, 250 (**e**, **g**); *ecfp*, 500; *wnt8*, 250 (**i–k**). Amounts of MO (ng/embryo): std MO and *ndst1* MO, 14. TR-Dx, 5 ng/embryo **g**. Amount of TOP-Flash reporter DNA, 100 pg/embryo. Scale bars, 20 μm

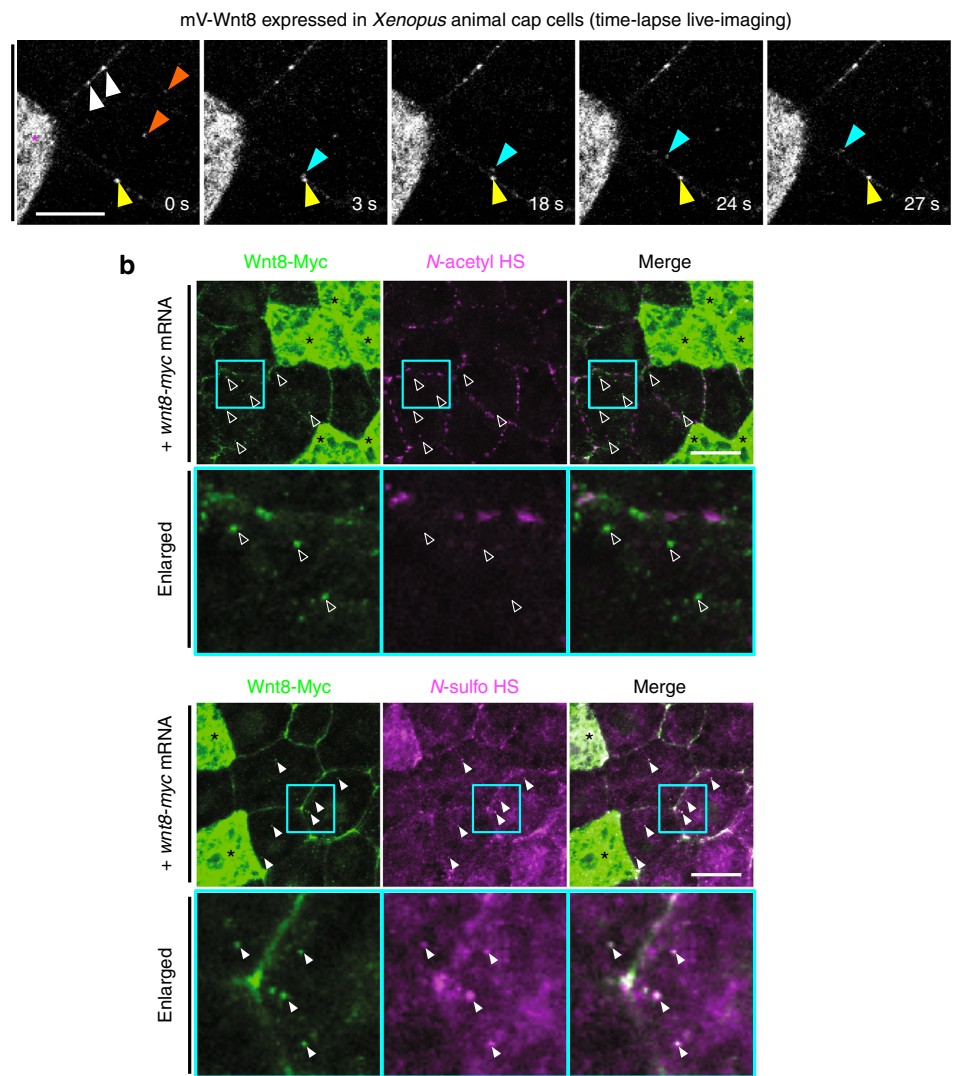

**Fig. 7** Involvement of the *N*-sulfo-rich HS clusters in Wnt8 internalization. **a** Time-lapse imaging of internalization of mV-Wnt8 puncta. Subapical region (basal to the tight junction) of the superficial cell of the *Xenopus* gastrula (st. 11.5) was observed. In the vicinity of mV-Wnt8 expressing cells (asterisk), puncta at the cell boundary (white and yellow arrowheads) and inside cells (orange arrowheads) were observed. The time point of each picture was as indicated. A budding and subsequently internalized punctum (tracked by a cyan arrowhead) was generated from a punctum (yellow arrowhead) on the cell membrane. See also Supplementary Movie 1. **b** Colocalization of internalized puncta of Wnt8 with *N*-sulfo-rich HS clusters. Wnt8-Myc expressing cells were as indicated (*), and the others were receiving cells. The puncta of Wnt8-Myc inside cells near Wnt8-Myc expressing cells are considered to be internalized, because the subapical region (basal to the tight junction) but not the apical cell surface was observed (arrowheads or open arrowheads). Upper panels, co-immunostaining of Wnt8-Myc and *N*-acetyl HS. Staining of *N*-acetyl HS was absent from internalized puncta of Wnt8-Myc (open arrowheads). Lower panels, co-immunostaining of Wnt8-Myc and *N*-sulfo HS. Colocalization of *N*-sulfo-rich HS clusters and Wnt8-Myc inside cells (arrowheads) was observed, suggesting involvement of *N*-sulfo-rich HS cluster in Wnt8 internalization. Images are a representative of at least two independent experiments. Amounts of injected mRNA (ng/embryo): *mV-wnt8*, 1.0; *wnt8-myc*, 0.25. Scale bars, 10 µm (**a**); 20 µm (**b**)

($r = 0.186$) (Fig. 9a). The preference of Frzb for *N*-acetyl-rich HS clusters was verified by several lines of evidence as follows. Firstly, overexpression of *ndst1* caused absence of mV-Frzb from the boundary between *ndst1*-overexpressing cells (Supplementary Fig. 13c), in which *N*-acetyl-rich HS clusters were shown to be reduced by Ndst1 (Fig. 3d). Secondary, mV-Frzb was accumulated on the cell surface of *gpc4*- but not *gpc4ΔHS*-overexpressing cells (magenta) to a higher degree than that of non-expressing control cells (Supplementary Fig. 13d), consistent with the observation that *N*-acetyl-rich HS clusters were increased by overexpression of *gpc4* (Fig. 4c). In addition, mV-Frzb was not accumulated more on the cell surface of *gpc5* or *gpc5ΔHS*-overexpressing cells than non-expressing control cells (Supplementary Fig. 13d), consistent with the observation that the amount of

*N*-acetyl-rich HS was not changed, though that of *N*-sulfo-rich HS was increased by overexpression of *gpc5* (Fig. 4c). These data concluded that Frzb preferentially associates with *N*-acetyl-rich HS clusters, in contrast to Wnt8, which prefers *N*-sulfo-rich HS clusters.

Consistent with endogenous Wnt8 (Fig. 2f–k), Wnt8-Myc preferentially colocalized with *N*-sulfo-rich HS clusters ($r = 0.780$) compared to *N*-acetyl-rich HS clusters ($r = -0.075$) (Fig. 9b). Remarkably, when Frzb was expressed in cells close to the *wnt8-myc* expressing cells, colocalization of Wnt8-Myc with *N*-sulfo-rich HS clusters was almost abolished ($r = 0.241$) and conversely that with *N*-acetyl-rich HS clusters was substantially increased ($r = 0.911$) (Fig. 9c), suggesting that Wnt8/Frzb complexes associate with *N*-acetyl-rich HS clusters. This shifting of the

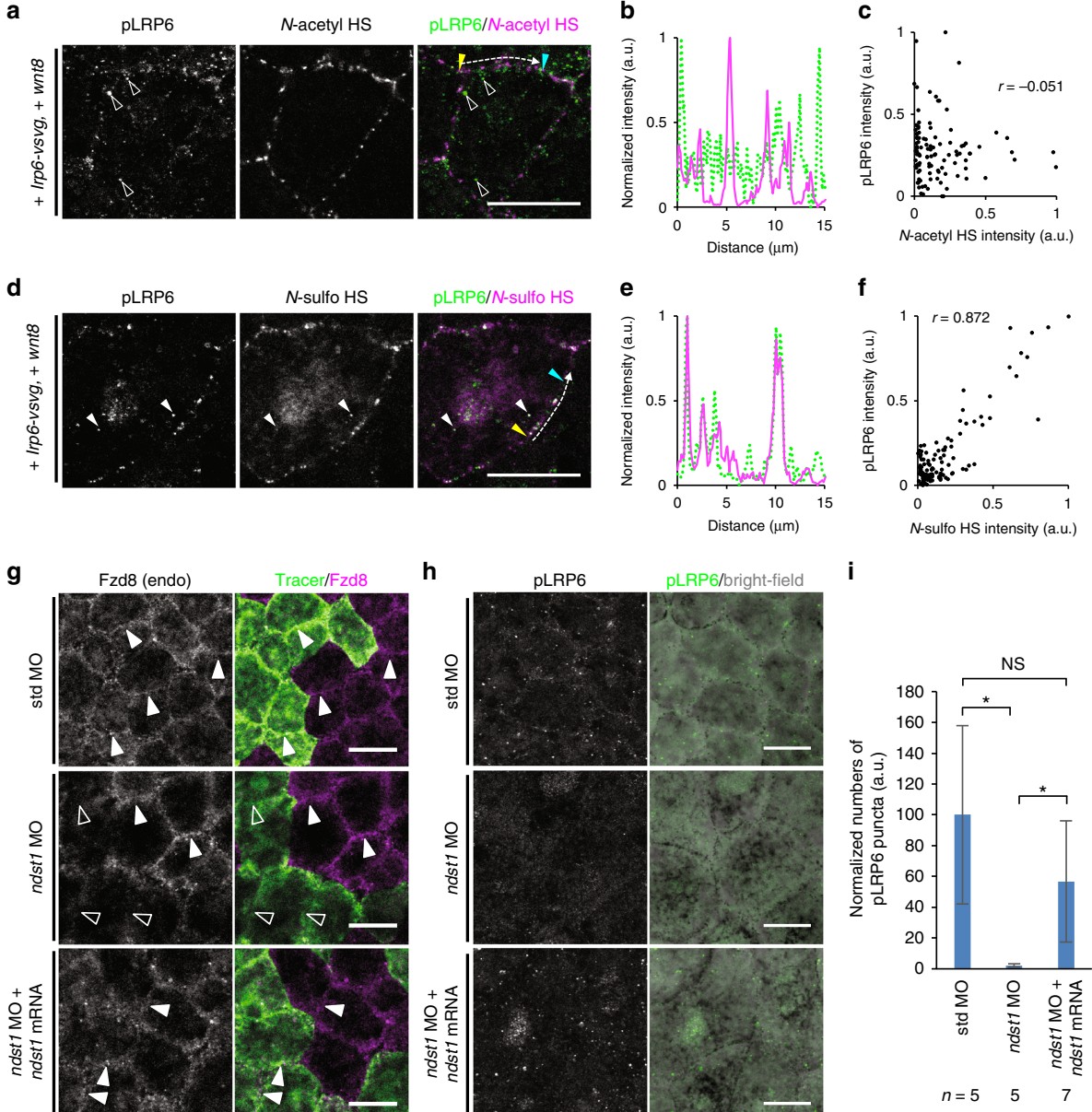

**Fig. 8** *N*-sulfo-rich HS clusters are involved in the formation of the Wnt/LRP6 signalosome. **a–f** The colocalization of phosphorylated LRP6 (pLRP6) and *N*-sulfo-rich HS clusters. Together with Alexa Fluor 647-dextran (tracer, not shown), mRNAs of *wnt8* and *lrp6-vsvg* were injected into the animal pole region of a ventral blastomere of 4- or 8-cell stage embryos. Signal intensities along white arrows were plotted (**b**, **e**), starting (yellow arrowheads) and ending (cyan arrowheads) points as indicated, respectively (**a**, **d**). Distributions of pLRP6 and *N*-acetyl HS were not correlated well (correlation coefficient *r* = −0.051), whereas distributions of Wnt8 and *N*-sulfo HS were highly correlated (*r* = 0.872). Note that pLRP6 staining is colocalized with *N*-sulfo HS staining on the membrane and also inside cells (white arrowheads) but not with *N*-acetyl HS staining (open arrowheads). **g** Requirement of *ndst1* for the membrane distribution of endogenous Fzd8. Experimental procedures are essentially the same as in Fig. 6c. Immunostaining was carried out with the anti-Fzd8. std MO, as a negative control, did not affect Fzd8 accumulation at the cell boundary (white arrowheads), whereas *ndst1* MO cell-autonomously reduced Fzd8 accumulation at the cell boundary (open arrowheads). Coinjection of *ndst1* mRNA recovered Fzd8 accumulation at the cell boundary (white arrowheads). **h** Requirement of *ndst1* for the phosphorylation of LRP6. mRNAs of *lrp6*, *wnt8*, and *ndst1* (for rescue experiment only, bottom) and indicated MO were injected into the animal pole region of a ventral blastomere of 4- or 8-cell stage embryos. Punctate staining of pLRP6 was impaired by *ndst1* MO, which was rescued by *ndst1* mRNA. **i** Quantification of pLRP6 puncta. Graph, mean±s.e.m. Significance level, **p < 0.01 (multiple comparison with pairwise Wilcoxon rank sum test). Numbers of analyzed images (*n*) are as indicated. See Methods for details. *Xenopus* embryos (st. 11.5) were fixed, stained, and analyzed. Images are a representative of at least two independent experiments. Amounts of injected mRNA (pg/embryo): *lrp6-vsvg*, 38; *wnt8*, 250; *ndst1*, 15 (**h**) or 6.7 (**g**). Amounts of injected MOs (ng/embryo): 28 (**g**), or 14 (**h**). Scale bars, 20 μm. a.u., arbitrary units

colocalization of Wnt8 with HS clusters was further confirmed by *ndst1* overexpression; while mV-Wnt8 alone accumulated on the *ndst1*-overexpressing cells, mV-Wnt8 together with Frzb was not localized on the cell boundary between two *ndst1* overexpressing cells (Fig. 10a). Consistent with the fact that *N*-acetyl-rich HS

clusters are not frequently internalized (Fig. 5a), intracellular puncta of mV-Frzb were much fewer than those of mV-Wnt8 (Fig. 10b, left and middle), but, in the presence of Frzb, the intracellular puncta of mV-Wnt8 were reduced (Fig. 10b, right). These data suggest that Frzb inhibits the internalization of Wnt8

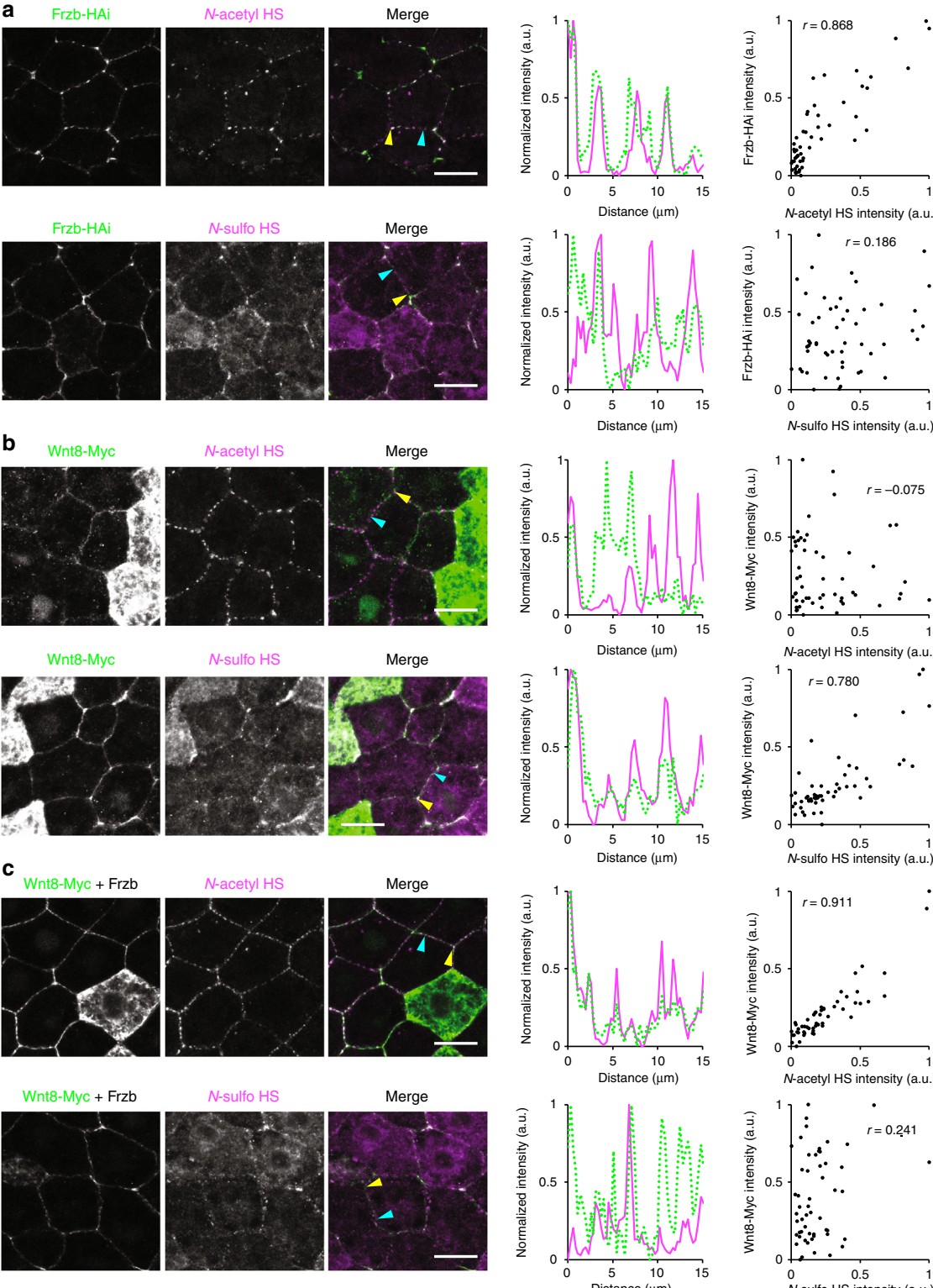

**Fig. 9** Frzb and the Wnt8–Frzb complex accumulates on *N*-acetyl-rich HS clusters. Intercellular distributions of HS and Wnt8 or Frzb in the *Xenopus* embryo (st. 10.5) were visualized by immunostaining (1st to 3rd panels from the left). Normalized intensities of Frzb-HAi or Wnt8-Myc (green) and HS (magenta) along the cell boundaries were plotted in graphs (4th panels). The start and end points are indicated by yellow and cyan arrowheads, respectively, in the merged pictures. Paired sets of normalized intensities of *N*-acetyl and *N*-sulfo signals at each position were plotted (5th panels). Values of calculated correlation coefficients (*r*) are shown in the graphs. **a** Frzb-HAi. Internally HA-tagged Frzb (Frzb-HAi) preferentially colocalized with *N*-acetyl HS compared to *N*-sulfo HS. **b** Wnt8-Myc. Conversely, Wnt8-Myc preferentially colocalized with *N*-sulfo HS compared to *N*-acetyl HS. **c** Wnt8-Myc in the presence of Frzb. When Frzb was expressed adjacently, Wnt8-Myc became colocalized preferentially to *N*-acetyl HS similar to Frzb-HAi. Images are a representative of four independent experiments. Amounts of mRNAs (ng/embryo): *frzb-HAi* 0.25; *wnt8-myc*, 0.25; for *frzb*, 1.0. Scale bars, 20 μm. a.u., arbitrary units

by shifting its association from *N*-sulfo-rich HS clusters to *N*-acetyl-rich HS clusters (Fig. 10c). This mechanism could account for the distribution range expansion of Wnt and the inhibition of Wnt signaling by Frzb[10, 15].

Accumulation of the Wnt8/Frzb complex on *N*-acetyl-rich HS clusters raised the question of whether Wnt8 on *N*-acetyl-rich HS clusters cannot transduce Wnt signaling. To answer this question, we used GPI-anchored Crescent (a member of sFRPs), Cres-GPI, to mimic the Frzb-*N*-acetyl-rich HS-glypican-GPI anchor con-furation[10], and to express Crescent cell-autonomously allowing for the comparison of the levels of nuclear β-catenin of Cres-GPI-expressing cells and non-expressing cells. The data showed that Cres-GPI-expressing cells accumulated Wnt8 but did not have an increased staining of nuclear β-catenin compared to adjacent non-expressing cells (Supplementary Fig. 13e), suggesting that Wnt8 that accumulated on *N*-acetyl-rich HS clusters via sFRP is non-functional for signal activation.

## Discussion

Starting from the finding that endogenous/exogenous Wnt8 is accumulated in a punctate manner on the cell boundary, we discovered new types of subcellular structures on the plasma membrane, *N*-sulfo-rich HS and *N*-acetyl-rich HS clusters. The distinct clustering of differently modified HS on the cell membrane in a single cell is a novel concept for elucidating cell biological organization and function of HS chains (Fig. 1c). The prevailing NA/NS domain model of HS explains differences in HS chains between tissues as differing ratios of NA and NS domains within individual HS chains (Fig. 1b). In our model, *N*-acetyl-rich HS and *N*-sulfo-rich HS chains can be considered as a variation of ratios of NA/NS domains, however, our model also describes that a single cell expresses different types of HS chains and that these HS chains form distinct clusters, neither of which is included in the NA/NS domain model (Fig. 1c).

*N*-acetyl-rich HS chains identified in this study might correspond to the previously reported "unsulfated heparan chains", which were biochemically identified in colon carcinoma cells as "aberrant" HSPGs, and whose biological roles other than as precursors of HS have not been considered[55]. However, in *Xenopus* gastrula, disaccharide analysis revealed that HexUA-GlcNAc (*N*-acetyl disaccharide unit) is also abundant[56]. We verified the presence of *N*-acetyl-rich HS chains in *Xenopus* embryos using "cellulose acetate membrane electrophoresis coupled with alcian immunostaining (CAME-AIS)", implying that unsulfated heparan chains may not be aberrant molecules, but rather may play a role, as exemplified with the distributions of Frzb and the Wnt8–Frzb complex (Figs. 9 and 10).

*N*-sulfo-rich and *N*-acetyl-rich HS clusters differ in several aspects, including core proteins, internalization tendencies, colocalization with Wnt signalosome components, and association preferences for Wnt and Frzb. These striking differences provide new insights into the molecular basis for morphogen behaviors in the extracellular space. Several models for the gradient formation of diffusible molecules have been proposed, including a restricted diffusion model[6] and a source-sink model[57]. In the source-sink model, endocytosis of ligands is required for a steady gradient of fibroblast growth factor (FGF), the sink of which is thought to be FGF receptors[57]. In the case of Wnt, *N*-sulfo-rich HS clusters fit the role of a sink more so than receptors, as *N*-sulfo-rich HS clusters are suggested to be internalized constitutively, independently of Wnt signaling (Fig. 5). Thus, our model of *N*-sulfo-rich HS clusters can unify the restricted diffusion and source-sink models for Wnt8. *N*-acetyl-rich HS clusters also contribute to restricted diffusion for Frzb (Fig. 10c), but tend to stay on the cell surface, acting much less

like a sink, to retain diffusible molecules on the cell surface, probably leading to the long-range distribution exemplified by Frzb and Wnt8–Frzb complexes[10]. Thus, *N*-acetyl-rich HS and *N*-sulfo-rich HS clusters constitute a cellular platform for the regulation of Wnt distribution and signaling together with Frzb (Fig. 10c).

Our data suggest that Gpc4 bears both *N*-acetyl-rich HS and *N*-sulfo-rich HS whereas Gpc5 bears *N*-sulfo-rich HS (Fig. 4b, c). Therefore, glypican core proteins appear to control the type of HS that is attached to them. Such core protein-dependent differences in GAGs have also been reported between syndecan-1 and -4 that were expressed in the same cell line[58]. Regarding the role of glypicans in Wnt signaling, the differences in HS chains between Gpc4 and Gpc5 may explain why Gpc4 in *Xenopus* has been implicated in non-canonical Wnt signaling but not in canonical Wnt signaling as analyzed by knockdown experiments[44]. On the other hand, knockdown of *Gpc4* suppresses both the canonical and non-canonical Wnt pathways in HEK293 cells[59]. This phenotypic variation may reflect combinations of expressed glypican genes. In addition, *Gpc3*, an ortholog of *Gpc5*, is reportedly involved in canonical Wnt signaling in hepatocellular carcinoma cells[60]. Consistent with our model (Fig. 10c), two glypicans in *Drosophila*, Dally and Dally-like protein (Dlp), have been suggested to play distinct roles in the distribution and signaling of Wg[6, 21]. That is, Dally acts as a classical co-receptor for local signaling, whereas Dlp has a role in both signaling and long-range transport[21, 22]. This previously rather puzzling dual role of Dlp can be accounted for by our model, if it is assumed that Dally (Gpc5 ortholog) forms *N*-sulfo-rich HS clusters, whereas Dlp (Gpc4 ortholog) forms both *N*-sulfo-rich and *N*-acetyl-rich HS clusters. Taken together with our data on the opposite roles of *N*-sulfo-rich and *N*-acetyl-rich HS clusters in internalization and signaling of Wnt8, it is reasonable to speculate that the *N*-sulfo-rich HS-containing portion of Dlp contributes to signaling, while the *N*-acetyl-rich HS-containing portion contributes to the retention of ligands for further transport. In addition, the greater internalization rate of Dally than that of Dlp[21] is consistent with our data on *N*-sulfo-rich HS (Gpc5 and Gpc4) and *N*-acetyl-rich HS (Gpc4) (Fig. 5). The contribution of Dlp to the long-range distribution of Wg raises the question of whether Wg binds to *N*-acetyl-rich HS unlike Wnt8, because *Drosophila* appears to lack sFRP genes[15]. Thus, our discovery of *N*-acetyl-rich and *N*-sulfo-rich HS clusters provides insights into HSPGs and their regulatory mechanisms regarding Wnt proteins.

Canonical Wnt signaling is considered to be initiated by signalosome formation[1, 2, 29, 51, 59, 61]. Our data suggest that *N*-sulfo-rich HS clusters serve as pre-existing cores to accumulate ligands, recruit the receptor Frizzled, and then phosphorylate LRP6 to form signalosomes (Fig. 10d). Of note, the basolateral distribution of Wnt8-Myc (Supplementary Fig. 2b) is similar to a reported distribution of endogenous phosphorylated LRP6 (pLRP6)[62]. HS clusters may also contribute to the filopodia-based transport of Wnt8a, because puncta of tagged Wnt8a proteins reportedly move along the filopodial protrusion[12, 13]. Furthermore, as implicated by the dual roles of Dlp discussed above (signaling and retention of Wg), the question of what kinds of pools of extracellular Wnt ligands contribute to signal activation remains a challenge in the field[6]. Our data suggest that some pools of Wnt ligands transduce signaling while others do not. Of note, some puncta of Wnt8 did not colocalize with *N*-sulfo HS and Fzd8 (Supplementary Fig. 11g, open arrowheads). One possible explanation is that HepSS-1 does not react with a certain type of *N*-sulfo HS that are modified by epimerization and *O*-sulfation[33]. These modifications of HS could also be involved in Wnt distribution and signaling. Indeed, analysis of *sulf-1*, which hydrolyzes 6-*O*-sulfation of HS, indicates that 6-*O*-sulfation enhances

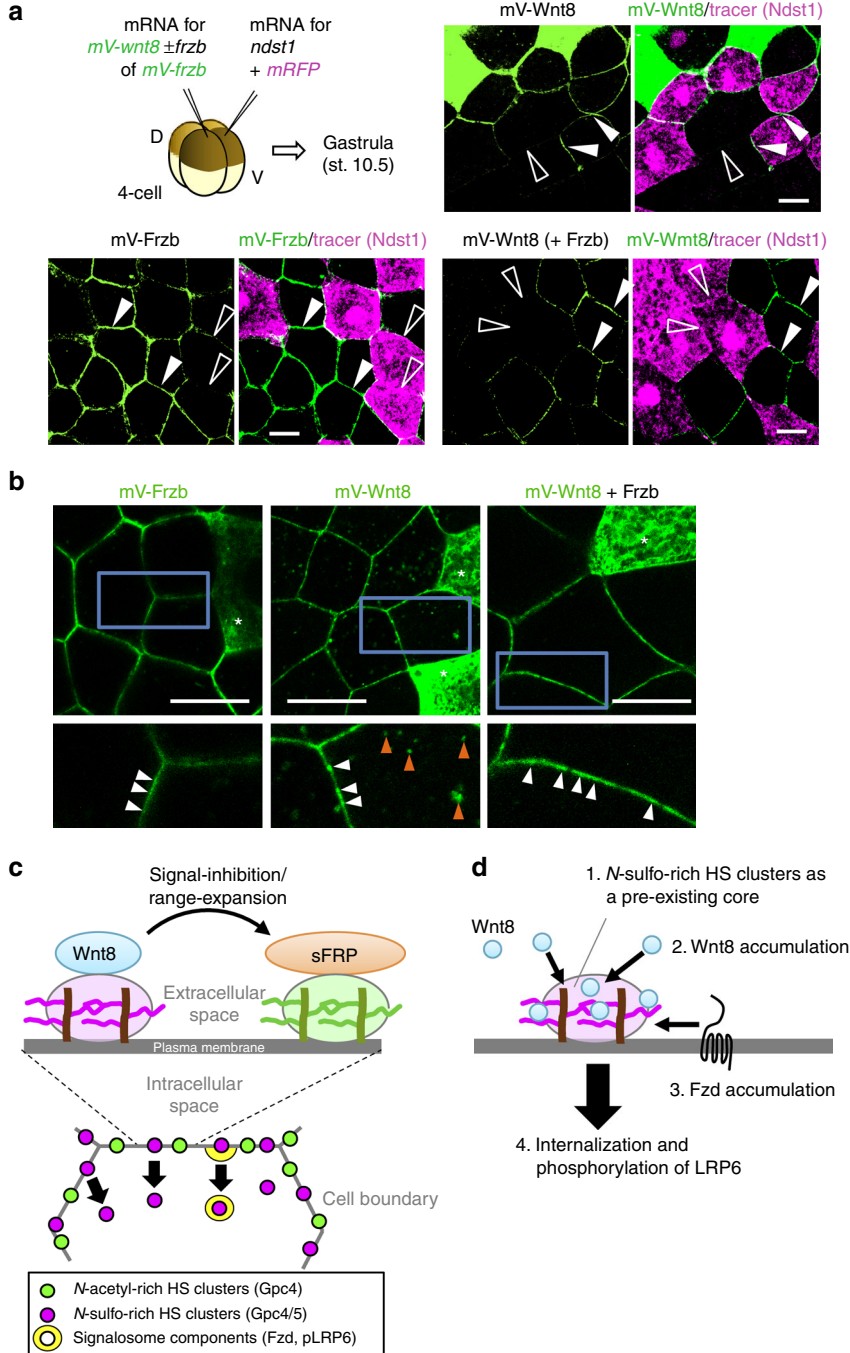

**Fig. 10** Frzb shifts Wnt8 from *N*-sulfo-rich to *N*-acetyl-rich HS clusters. **a** Frzb inhibits the accumulation of Wnt8 on *ndst1* overexpressing cells. Experimental procedures were as illustrated. Although mV-Wnt8 was accumulated on *ndst1*-overexpressing cells (upper right), mV-Frzb was not accumulated on *ndst1*-overexpressing cells (lower left). When coexpressed with Frzb, mV-Wnt8 was not accumulated on *ndst1* overexpressing cells similar to Frzb (lower right). **b** Frzb inhibits internalization of Wnt8. Left panels, the characteristic puncta of mV-Frzb (asterisks, source cells of mV-Frzb). Middle panels, mV-Wnt8 in the intercellular space (white arrowheads) and inside the cells (orange arrowheads) (asterisks, source cells of mV-Wnt8). Right panels, in the presence of Frzb, punctate staining of mV-Wnt8 (asterisks, source cells of mV-Wnt8) became smoother in the intercellular space (white arrowheads) and was barely detected inside the cell. Lower panels, magnified images of the boxed regions in the upper panels. *mV-wnt8* and *frzb* mRNAs were separately injected into different blastomeres. **c** A model of the *N*-sulfo-rich and *N*-acetyl-rich HS cluster system for the regulation of the extracellular distribution and signal reception of the Wnt morphogen. **d** A model for the initiation of signalosome formation. *N*-sulfo-rich HS clusters (magenta) work as a pre-existing core to accumulate Wnt8 (green) prior to the recruitment of the Fzd receptor and following phosphorylation of LRP6. Images are a representative of at least two independent experiments. Amounts of mRNAs (ng/embryo): *mV-wnt8*, 1.0 (**a**) or 2.0 (**b**); *mV-frzb*, 0.39 (**a**) or 2.0 (**b**); *frzb*, 0.48 (**a**) or 2.5 (**b**); *ndst1*, 0.5. Scale bars, 20 μm. a.u., arbitrary units

binding of Wnt to HS but downregulates Wnt signalling[63]. Thus, our data suggest that only the portion of Wnt8 associated with *N*-sulfo-rich HS serves as the pool that directly activates Wnt signaling.

Thus, the opposite roles of *N*-sulfo-rich and *N*-acetyl-rich HS clusters provide a new model for the distribution and gradient formation of secreted proteins, as well as the initiation and regulation of Wnt signaling, and also shed light on the subcellular organization of HS proteoglycans on the plasma membrane.

## Methods

**Xenopus embryo manipulation and microinjection.** All experiments using *Xenopus laevis* were approved by The Office for Life Science Research Ethics and Safety, University of Tokyo or The Institutional Animal Care and Use Committee, National Institutes of Natural Sciences. Manipulation of *X. laevis* embryos and microinjection experiments were carried out according to standard methods[10, 64] as follows. Briefly, unfertilized eggs were obtained from female frogs injected with gonadotropin (ASKA Pharmaceutical), and artificially fertilized with testis homogenate. Fertilized eggs were dejellied with 2% L-cysteine-HCl solution (pH 7.8), and incubated in 1/10× Steinberg's solution at 14–20 °C. Embryos were staged according to Nieuwkoop and Faber[65]. Synthetic mRNAs were transcribed from plasmid DNAs using mMessage mMachine SP6 kit (Ambion) and microinjected into early (2–16 cells) embryos. The amounts of injected mRNAs are described in the Figure legends. For tracing MO-injected cells, 2.5–5.0 ng of FITC-dextran (Molecular Probes, D1820) was coinjected. Heparitinase (25 µU, #100703, Seikagaku Biobusiness, a mixture of heparitinase I and II at 4:1 ratio) was microinjected into the blastocoel of embryos at stage 6.5 (48 cells)[66]. Similarly, blastocoel injection was carried out for PI-PLC (800 µU, Molecular Probes, P6466), anti-HS antibodies or BSA-AF647.

**Antibodies.** Rabbit polyclonal anti-Wnt8 antibody was generated with purified recombinant *X. laevis* Wnt8 protein (residue 44–358) expressed in *E. coli*. Pre-immune serum was obtained from the same rabbit just before immunization. Other primary antibodies used for immunostaining were as follows: mouse monoclonal anti-Myc (9E10, BIOMOL, 1:1000), anti-Myc (4A6, Upstate, 1:2000), anti-HA (12CA5, Roche, 1:1000), anti-*N*-acetyl HS antibody (NAH46, Seikagaku, 1:200 or in-house preparation, 1:50), anti-*N*-sulfo HS (HepSS-1, Seikagaku, 1:200 or in-house preparation, 1:400), anti-β-tubulin (TUB2.1, Sigma, 1:200) antibodies; rabbit polyclonal anti-pLRP6 (2568S, CST, 1:100–200), anti-HA (561, MBL, 1:200), anti-Myc (562, MBL, 1:200), anti-ZO-1 (Invitrogen, 1:200), and anti-β-catenin (C2206, Sigma, 1:1000) antibodies; and goat polyclonal anti-Fzd8 antibodies (Novus Biologicals, 1:100–200). Secondary antibodies used were goat or donkey polyclonal anti-mouse or rabbit IgM or IgG antibodies labeled with Alexa Fluor 488, Alexa Fluor 546, Alexa Fluor 555 or Alexa Fluor 647 (1:500–1000, Molecular Probes). For double-color fluorescence immunostaining for *N*-acetyl HS ([GlcA–GlcNAc]ₙ) and *N*-sulfo HS ([GlcA–GlcNS]ₙ) using mouse IgM monoclonal antibodies, NAH46 and HepSS-1 (Fig. 3a), anti-*N*-acetyl HS (NAH46) and anti-*N*-sulfo HS (HepSS-1) antibodies were directly labeled with Alexa Fluor 488 5-SDP ester (Molecular Probes, A30052) and Alexa Fluor 647 NHS ester (Molecular Probes, A20006), respectively as follows. Conjugation reaction was performed by mixing antibody solution (1.0 mg/ml in PBS) and 1/50 volume of the reactive dye (10 mg/ml) and incubating for 1.5 h at room temperature under dark conditions. Unconjugated dye was removed by gel filtration using Bio-Spin 6 Tris Columns (#732-6227; Bio-Rad) to collect the labeled antibody as a flow-through fraction.

**Western blotting.** Western blotting was performed with PVDF membranes and HRP-conjugated secondary antibody. Bound secondary antibody was detected using chemiluminescence (Chemi-Lumi One Super, Nakarai Tesque) with X-ray films (Fujifilm). All uncropped western blot images can be found in Supplementary Fig. 14.

**Morpholino oligos.** Because *X. laevis* is an allotetraploid, we used an 1:1 mixture of two MOs targeting transcripts from both homoeologs as follows: wnt8 (the same as wnt8a), GAACAAAGTGGTGTTTTGCATGATG[67] and CGCTTTTT CATGGTGAAGGCTGATAT; ndst1, AGGAATGGCACAAGCTCACAAATGC and AGGAGTGGCACAAGCTCACAAATGC; gpc4, TGCATGGTGAGAA CGAAAGCGATC[44] and AAAAGGAGATCCAAAGCATGGTGAT; gpc5, ACAGGAACCGGCTCCTTGTGCCTCT and ACAAAGCCCTGTACACATGTA-TATC; fzd8, GTCACTGTCTGAGGCTCATTCCTGG[68] and GCAGCGACAGA-TACGGACACTCCAT[69]. MO sequences without reference were newly designed in this study. Amounts of MOs indicated in Figure legends are the total amounts of an 1:1 mixture of the two MOs. std MO (CCTCTTACCTCAGTTACAATTTATA) was used as a negative control.

**Plasmid construction.** pCSf107mT[10] was used as a vector possessing 4× SP6/T7 transcription terminator sequences to make plasmid constructs for mRNA synthesis unless otherwise mentioned. pCSf107-ndst1 and pCSf107-gpc5 were constructed by inserting a PCR product for the full length coding sequence (CDS) of *X. laevis* ndst1 and gpc5, respectively, amplified with a primer set (ndst1, ggaattccACCATGAGCTTTGTCCCTGAAG and gctctagatTACCTAGTGT TCTGAAGCTCC; gpc5, ggaattccaccATGTCCCGTTCGCTCTCGCT and gctcta-gaTTAGATATACTGACATATGA) from the Blumberg library into pCSf107mT using the EcoRI/XbaI sites. Gpc4ΔHS mutant[44] was constructed as using PCR-based mutagenesis. Gpc5ΔHS mutant was constructed by introducing point mutations changing Ser499, Ser511 and Ser523 at putative HS attachment sites to Ala using PCR-based mutagenesis and the In-Fusion system (Clontech). Primers used for this mutagenesis are as follows: AGTTTTTCATTTCCTCCAGCCCCTT, AGGAAATGAAAAACTGGAAGACGCAGATCCCgcTGGTGAC, TTTTGT GCCTGTGTGCGTAGAAGAATCCCCAGCACCTTGG, CACACAGGCA CAAAACTAATACCTG.

Golgi-tethered *Bacillus cereus* PI-PLC (Golgi-PI-PLC) was constructed by connecting in-frame the Golgi-directed signal peptide and transmembrane domain (1–122 aa) of *homo sapiens* Galnt3 (GenBank accession number X92689) and the synthetic *B. cereus* CDS sequence without the signal peptide (25–329 aa), codons of which were synonymously changed according to the *Xenopus* codon usage (FASMAC, the sequence was deposited to Genbank). C-terminally mV-tagged Wnt8 and Frzb were constructed as follows. pCSf107SPw-mT and pCSf107SPf-mT were constructed, which encode the signal peptides of Wnt8 and Frzb, respectively. The mV CDS was inserted into the BamHI site of pCSf107SPw-mT and pCSf107SPf-mT to construct pCSf107SPw-mV-T and pCSf107SPf-mV-T, respectively. Then a CDS of Wnt8 or Frzb without signal peptides[10] was inserted into pCSf107SPw-mV-T or pCSf107SPf-mV-T to construct pCSf107SPw-mV-Wnt8-T or pCSf107SPf- mV-Frzb-T, respectively. pCS2-Cres-GPI[10] was used as a template for expressing Cres-HA-GPI. pCSf107-SP-mECFP-GPI was constructed using pCSf107mT by inserting PCR products of the signal peptide of TGFβ1 (sp), mECFP (monomeric ECFP) and the GPI-anchor signal, which were amplified with the following primer sets, respectively:

 sp: gaagatctATGCCGCCCTCCGGGCTGCG and
 cgttcgaaatcgatggatccGGCGGGCCGGCCGGCTAGG;
 mECFP: cgttcgaaGTGAGCAAGGGCGAGGAG and
 cgacgcgtCTTGTACAGCTCGTCCATGC;
 GPI: cgacgcgtctcgagtctagaCTTGAGGAAACAACCCCAAATAA and
 gactagtCTAAGTCAGCAAGCCCATGG.

**Luciferase reporter assays.** Luciferase reporter assays with the TOP-Flash reporter[45] were carried out as follows. mRNAs, MO, and the TOP-Flash reporter was injected into the animal pole region of a ventral blastomere at 4- or 8-cell stage *Xenopus* embryos. Injected embryos were collected at st. 11.5. A pool of three embryos was counted as one sample (e.g., n = 8 corresponds to 24 embryos).

**Statistical analyses.** Sample sizes were empirically determined. For comparison of a single pair of quantitative data, Student's or Welch's *t*-test (two-sided) was carried out after comparing the variances of a set of data by *F*-test. We used this parametric test only when the normal distribution could be assumed. Multiple comparisons were carried out with pairwise Wilcoxon rank sum test (two-sided) in which significance levels were adjusted by the Holm method using R. Neither randomization nor blinding was used in this study.

**Cell culture and DNA transfection.** HeLa cells were cultured in Dulbecco's Modified Eagle Medium supplemented with 10% fetal bovine serum (FBS) with or without penicillin–streptomycin (100 U/ml and 100 µg/ml, respectively) in a 5% $CO_2$ incubator at 37 °C. HeLa cells were provided by Dr. Shin-Ichi Osada or Dr. H. Hikasa. Neither authentication nor the test for mycoplasma was performed. For immunohistochemistry, cells were cultured on glass coverslips in 12-well dish on glass coverslips and transfected with the following amounts of DNA per well: 200 ng hLRP6, 50 ng Mesd, 2 ng mFz8/mFz8-mVenus, 15 ng hAxin-Myc, 8 ng hGSK3β-FLAG, with the transfection reagent, PEI-Max. Cells were harvested >19 h post transfection and treated with Wnt3a or control conditioned medium for 2 h at around 60% confluency.

**Immunostaining.** *Xenopus* gastrula embryos were fixed with MEMFA (0.1 M MOPS, pH 7.4, 2 mM EGTA, 1 mM $MgSO_4$, 3.7% formaldehyde) and immunostained by standard protocols with Tris-buffered saline[64]. For permeabilization, 0.1% Triton X-100 was used. For immunostaining for *N*-acetyl or *N*-sulfo HS and endogenous proteins, such as Wnt8 and Fzd8, secondary antibodies were pre-adsorbed with acetone powder of fixed *Xenopus* gastrula, to reduce background staining. For double staining of *N*-acetyl and *N*-sulfo HS of *Xenopus* embryos, the two directly labeled antibodies described above were applied to fixed embryos at once. Fixed embryos were reacted with two primary or two secondary antibodies at once because no cross-reactions were observed. HeLa cells were fixed with 4% PFA/PBS for 15 min at RT. For double staining of *N*-acetyl and *N*-sulfo HS of HeLa cells (Supplementary Fig. 3f), fixed cells were incubated with HepSS-1 antibody followed by reaction with anti-mouse IgM secondary antibody conjugated with Alexa Fluor

546, and re-fixed with MEMFA. Then, the cells were reacted with NAH46-Alexa Fluor 488 at the dilution of 1:200. For single staining of b-tubulin, *N*-acetyl HS, or *N*-sulfo HS in Supplementary Fig. 3g, fixed cells were permeabilized with 0.5% Triton X-100 for 10–15 min at room temperature, or not permeabilized. Then cells were incubated with primary antibodies for 10–15 min at room temperature. Otherwise, prior to antibody incubation, cells were permeabilized with TBST (TBS with 0.01% Tween20) for 10 min (for Axin staining) or 40 min (for the staining of the others) at RT or not permeabilized (Supplementary Fig. 13a, b, d), and blocked with 2% BSA in TBS for 1 h at RT. The specimens were incubated with the following primary antibodies for 1 h at RT: anti-*N*-acetyl HS (NAH46, in-house preparation, 1:50), anti-*N*-sulfo HS (HepSS-1, Seikagaku, 1:200), anti-GFP (A11122, Invitrogen, 1:5000), anti-caveolin (sc-7875, Santa Cruz Biotechnology, 1:50), diluted with TBS. Following this, the samples were incubated with the secondary antibodies for 1 h at RT: anti-rabbit or mouse Alexa 488 or 555 antibody (Invitrogen, 1:500). Washing solution was TBS.

**STED microscopy.** STED observation was performed according to Fukata et al.[35] using a TCS SP5II STED CW system (HCX PL APO ×100, NA 1.4 oil immersion objective; Leica). Because wildtype embryos were destroyed by STED observation due to heat generation in pigment granules, albino embryos were used for analysis, which were fixed at st. 11.5 and stained with Alexa Fluor 488-labeled secondary antibody. Obtained STED images were processed with the built-in deconvolution algorithms of the LAS-AF software (Leica). FWHM (full width half maxima) of each HS cluster was semi-automatically quantified using ImageJ with "FWHM line" plug-in (Dr. Lim Soon Yew John, in ImageJ Mailing List).

**Purification of GAGs from *Xenopus* embryos and standard GAGs.** GAGs were purified from 400 embryos of the *Xenopus* gastrula (st. 11.5) according to a standard method[70]. *E. coli* K5 derived heparosan (OrigoTech, GAG070) and two types of chemically modified HS derived from bovine kidney in "heparan sulfate chemically modified kit" (Seikagaku, 400646), completely de-sulfated, *N*-acetylated (CDSNAc) HS and completely de-sulfated, *N*-sulfated (CDSNS) HS (see Supplementary Fig. 5b for their structures), were used as standards of GAGs.

**Alcian-immunostain of GAGs coupled with HS electrophoresis.** Cellulose acetate membrane electrophoresis of GAGs was carried out according to a previous report[36] with minor modifications to improve separation. Briefly, 0.1 M pyridine/0.47 M formic acid (pH 3) buffer and a cellulose acetate membrane (SELECA-VSP, Toyo Roshi Kaisha, Ltd.) were used. To prevent evaporation of the solvent from the membrane, which causes a flow toward the center and hence poor separation of CDSNAc HS and CDSNS HS, the solvent level of the cathode side was kept at least 12 mm higher than that of the anode side (Supplementary Fig. 5a). Electrophoresis was carried out in a constant current condition (1 mA/cm) for 80 min. The membrane was then stained with 0.1% alcian blue/0.1% acetic acid solution for 10 min, washed with water, and dried prior to imaging. To immunostain GAGs on the cellulose acetate membrane after the electrophoresis, we developed a method named CAME coupled with "alcian-immunostaining (AIS)" (CAME-AIS), as follows. After the electrophoresis, the membrane was briefly soaked in 0.1% alcian blue/0.1% acetic acid solution for 15 s, then quickly moved to a tray with water and washed three times with water (not exceeding 15 min in total). This alcian blue staining process serves as both staining and immobilization of GAGs on the membrane, and therefore should be done very quickly, otherwise the following immunostaining will not work; that is, completely stained GAGs were not recognized by antibodies. The AIS procedure was similar to a standard western blotting. Briefly, the membrane was blocked with 5% skim milk/TBS at 4 °C overnight, then incubated with primary antibody (NAH46 or HepSS-1) solution diluted with Can Get Signal Immunoreaction Enhancer Solution 1 (Toyobo) for 2 h at room temperature, followed by washing with 0.05% Tween 20/TBS three times and with TBS once. The membrane was then incubated with HRP-conjugated secondary antibody (anti-mouse IgM-HRP, Abcam, ab97230) solution diluted with Can Get Signal Immunoreaction Enhancer Solution 2 (Toyobo) for 2 h at room temperature, followed by washing with 0.05% Tween 20/TBS three times and with TBS once. Bound secondary antibody was detected using chemiluminescence (Chemi-Lumi One Super, Nakarai Tescue) with X-ray films (Fujifilm). Images of the films for AIS and the membrane for alcian blue staining were digitized with a scanner. All uncropped scanned images can be found in Supplementary Fig. 5d-f.

**Fluorescence image acquisition.** Image acquisition was performed using an LSM710 system (×40, NA 1.2 water immersion objective or ×63, NA 1.4 oil immersion objective; Zeiss), an LSM5 PASCAL system (×63 NA 1.2 and ×20 NA 0.60; Zeiss), an A1Rsi system (×63 NA 1.3 water immersion objective; Nikon) or a TSC SP8 system (×40, NA 1.10 water immersion objective; Leica). Bright field images were acquired by detecting transmitted light (488 or 552 nm). For live-imaging, *Xenopus* gastrula embryos were embedded with 1.5% LMP agarose (#16520-050; Invitrogen) gel in 1/10× Steinberg's solution on 35 mm glass-based dishes (Iwaki) or mounted in a house-made imaging chamber. For confocal imaging of immunostaining, stained embryos were mounted in shallow wells on 2%

agarose plate, flattened with a coverslip. Images were cropped and/or processed with Photoshop CS4 (Adobe) or ImageJ (NIH). Fluorescent intensity was measured using Image J (NIH) or Zen (Zeiss).

**Image analyses.** For counting pLRP6 puncta (Fig. 8i), the "Analyze Particles" function of ImageJ was used. The threshold was determined by the "MaxEntropy" algorithm. Particles whose area were not less than two pixels were counted. Overlapping of images obtained from double staining (Supplementary Fig. 11g) were calculated using ImageJ, as follows. Fluorescence images were binarized with a threshold determined by the "Otsu" algorithm for HepSS-1 staining or the "Triangle" algorithm for Fzd8 staining, then overlap images were calculated as the intersection of the binarized images, using the "AND" operation of the "Image Calculator" function.

**Data availability.** The authors declare that all data supporting the findings of this study are available within the article and its Supplementary Information files or from the corresponding author upon reasonable request.

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

## Acknowledgements

We thank A. Miyawaki for Venus cDNA; N. Ueno for Gpc4 cDNA; E.M. De Robertis for internally tagged Frzb-HA cDNA; R. Moon for pCS2+ Xwnt8; C. Niehrs for mMesd, mFz8, hGSK3β-FLAG; X. He for hLRP6; H. Hikasa for mAxin-Myc; T. Nakamura and Seikagaku Corp. for NAH46 antibody and hybridoma; O. Yoshie for HepSS-1 hybridoma; K. Nishiguchi for technical assistance for generation of Wnt8 antibody; H. Takeda for Zeiss LSM710 confocal microscopy; Y. Fukata and M. Fukata for STED microscopy; Spectrography and Bioimaging Facility, NIBB Core Research Facilities, for Nikon A1Rsi confocal microscopy; K. Itoh and S. Sokol, T. Yabe, T. Suzuki, K. Kawade and M. Suzuki for critical comments and discussion; C. Kawaguchi and G. St. Clair for editing the manuscript; M. Kondo, R. Harland and S. Hayashi for critical reading of the manuscript. This work was supported in part by MEXT/JSPS KAKENHI (24870031 and 15K14532 to Y.M.; 24111002 to S.T.; 25251026 to M.T.), a research grant from the Center for the Promotion of Integrated Science of SOKENDAI to Y.M. and a research grant from Daiko Foundation to S.T. Y.M. was a JSPS Research Fellow.

## Author contributions

Experimental contributions were as follows: experiments with *Xenopus* (Y.M. and T.Y.) and cultured cells (T.Y. and Y.M.); generation of anti-Wnt8 antibody (R.T. and Y.M.); analyses of endogenous Wnt8 (Y.M.); analyses of *ndst1* (Y.M. and T.Y.); identification and characterization of glypicans as core proteins for *N*-sulfo/*N*-acetyl-rich HS clusters (T.Y. and Y.M.); biochemical analyses of HS (Y.M., and S.M.); hybridoma culture and production of antibodies (M.M.). The project was mainly conceived by Y.M., T.Y., S.T. and M.T. S.Y. supervised biochemical analyses of HS. The paper was written by Y.M., T.Y., S.T. and M.T.

## Additional information

**Competing interests:** The authors declare no competing financial interests.

