## [Peer Review File · Nature Communications]

Reviewers' Comments:

Reviewer #1 (Remarks to the Author)

Mii et al.

This interesting manuscript addresses the regulation of Wnt8 in *Xenopus*, with particular emphasis on the roles of heparan sulfate proteoglycans, and exogenous inhibitor, Frzb. The authors utilize two HS-specific antibodies, both IgMs, to propose that there are discrete populations of HS on cell surfaces, one sulphate-rich the other sulphate poor, corresponding with the known specificities of the antibodies. They present data to suggest that sulphate-rich pools, on glypicans, have internalization/signaling potential, while the sulphate –poor pool, also on a glypican, does not. Moreover, Frzb, a possible soluble inhibitor of Wnt localizes to the sulphate-poor HS clusters and sequesters Wnt8 and thereby diminishes signaling.

These are interesting concepts, supported by high quality micrographs, and there is much of interest to specialists and perhaps others, but it appears that in many places the data are over-interpreted. Some areas of concern are as follows:

1. The authors contend that they have identified an entirely new paradigm for HS structures, and propose that the sulphate-poor clusters represent unsulfated heparosan. This is distinct from the high sulphated clusters. While the authors may have shown that HS with different sulfation patterns exist in a single tissue (a potentially novel and interesting finding), that is as far as it goes without characterizing the chains. The NAH46 antibody was raised against K5 polysaccharide (which is unsulfated), but the fact that it recognizes vertebrate HS does not show the presence of totally unsulfated chains. Possibly the unsulfated (NA) domains are larger or more numerous than in other HS pools. Moreover, the authors' own data shows that the high-sulfated pool, recognized by the HepSS-1 antibody, also contains NA domains. This is because the authors show sensitivity of the high-sulfated pool to heparitinase, an enzyme that only hydrolyzes NA domains. The authors also cannot rule out that the selective ability of the two antibodies to bind different HS pools is simply steric, since both are large IgM molecules. Short of purifying and characterizing the chains, the authors cannot justify their claim of an entirely new concept in HS synthesis, particularly when there is so much evidence to the contrary in the literature. Chains of varying fine structure are well known, that may be all that is discriminated here.
2. The authors make much of "internalized" Wnt, but it is very unclear how they know this. The methods are not sufficiently detailed to be sure. Puncta that are not at the cell margins (cell-cell interaction sites) are not necessarily internal. In addition, "signalosomes" are identified through Dvl2/Fzd8 codistribution. However, staining for Dvl2 is very sparse, and given the extensive Fzd8 staining in many cases, specificity of the codistribution patterns is not entirely convincing. If the authors wish to prove interaction here and elsewhere, then well-known FRET methods would have to be used. There are similar concerns with the beta-catenin data, nuclear staining, where present, is far from uniform and generally unconvincing.
3. An important starting point for this data is the codistribution of endogenous Wnt8 with high-sulfated HS (Fig 1H for example). However, a very significant portion of the Wnt8 is not apparently colocalized with the HS pattern. If the Wnt8 is lost by NDST1 MO (Fig 2D but needs careful quantitation), then there are quite possibly HS pools not detected by the HepSS-1 antibody. The authors seem to not take this possibility into account.
4. The authors use fluorescence microscopy almost entirely, but this mostly lacks any quantitation and there are places where biochemical techniques should be used to confirm the imaging data. What affinities does Wnt8 have for HS, does it require high-sulfated (NS domains) of HS? Does Frzb have high affinity for NA domains of HS? It is difficult to imagine that Wnt8 can transfer from NS-rich HS to NS-poor HS since the former will almost certainly have higher affinity. Presumably Frzb competes for the Wnt8, which is itself enriched at sites of high NA content (nicely shown and potentially important but see point 5) and this mediates the transfer of Wnt8?
5. Figure 5. Why is the staining pattern for N-sulfo HS so different here compared, for example,

with Fig 2? Is it likely that over-expression of Frzb disturbs the distribution of N-sulfo HS and indeed may be lost from the cell surface? If so, this introduces a potential problem of interpretation. The idea that Frzb localizes to a non-signaling compartment of HS is intriguing and very interesting, but there needs to be caution applied here to data interpretation.

6. The STED microscopy, while a current technique, seems not to add much. The authors note a difference in dimensions between the sulphate-rich and sulphate-poor HS clusters. However, they should perhaps note that this difference is in the order of the diameter of an IgM antibody (35-45nm) and so may not be informative.

7. Supplementary Fig 8. Clearly much of the over-expressed Wnt8 and Fzd8 does not colocalize with the N-sulfo clusters. Moreover it seems clear, by comparison of Figs a and b, that over-expression of Wnt8 alters the nature of the Fzd staining considerably. This does not negate the very nice work on endogenous Wnt8 but shows that caution should be applied to data interpretation.

8. A minor point. In the very useful experiments to deplete either GPC4 or 5, does depletion of one affect the expression of the other? This may have bearing on the results.

Reviewer #2 (Remarks to the Author)

This study aims to understand the distribution of Wnt signaling proteins *in vivo*, in *Xenopus* embryos. Using a newly generated antibody to Wnt8, the authors show the graded distribution of the protein to membrane puncta and their partial colocalization with clustered heparan sulfate proteoglycans (HSPGs). The Wnt8 protein was only associated with the N-sulfo HSPG molecules, one of the two major modifications of HSPG chains. Finally, the authors demonstrate that the N-sulfo clusters are frequently internalized and play a positive role in Wnt signaling. By contrast, N-acetyl HSPG accumulate Frzb, a secreted Wnt antagonist. The authors further show that glypican 5 is a major core protein for the N-sulfo clusters, whereas glypican 4 contributes to both N-sulfo and N-acetyl clusters. Based on the colocalization of the Wnt8-N-sulfo clusters with Frizzled and Dishevelled, the authors propose that the HSPG clusters represent predefined sites for signalosome formation.

Overall, the reported findings reflect a significant effort, are highly original and would be of interest to the field. The visualization of N-acetyl- and N-sulfated (N-sulfo) modifications of HSPGs as separate clusters *in vivo* has never been reported. Similarly, a role of HSPGs in Wnt signaling has been known, but the association of a Wnt protein with a specific form of HSPG is a significant advance corroborated in this study by the knockdown of the relevant NSDT enzyme and glypicans. The majority of the experiments include relevant controls for antibody and morpholino specificity. There is no doubt that these findings should be published, however, the remaining question is whether the authors' conclusions are sufficiently supported by data. The authors are encouraged to take into account the following two comments.

1. The work would be strengthened by biochemical evidence for the existence of separable 'N-sulfo' and 'N-acetyl' clusters and the Wnt8-N-sulfo complexes. This could be done in pull down assays with the same antibodies that were used for immunohistochemistry.

2. The proposed role of N-sulpho HS in Wnt signalosome formation can be further confirmed by monitoring the distribution of Wnt8 protein added to HeLa cells. This is an independent system, in which the N-sulpho and N-acetyl HS clusters are differentially distributed (this study) and the signalosomes were first characterized (Bilic et al., 2007). Signalosome formation can be corroborated by the appearance of a specific modification, such as LRP6 phosphorylation.

Other points

--The signaling assays are based on embryonic phenotypes and beta-catenin localization, but proper quantification is lacking.

-- Fig. 1H, Fig.4C, Fig S3E, Fig S8a. The conclusions are based on puncta colocalization, which is

sometimes difficult to see.

--Fig S4B. To exclude a nonspecific effect of the NDST1 morpholino on RNA translation, Wnt8 levels can be shown.

--Does overexpressed Ndst1 (alone or together with Wnt8) increase TOP-Flash activity?

--Is the phenotype of the Ndst1-knockout mice (Ringvall et al., 2000, JBC; Grobe et al., 2005; Development) consistent with the proposed role of this enzyme in Wnt signaling?

Reviewer #3 (Remarks to the Author)

The Wnt signalling pathway is one of the most important signalling pathways during development and tissue homeostasis. It regulates cell proliferation, stem cell maintenance, cell differentiation and regeneration of various organs. During development, Wnt/beta-catenin signalling is important organizing the embryonic body by setting the anterioposterior. Therefore, Wnt falls into the class of form-giving substances – so-called morphogens. To fulfil its morphogenetic function it is released from a localised source and forms a long-range signalling gradient. However, the endogenous Wnt protein gradient has not been shown in vertebrates so far. Mii and colleagues start with introducing an immune-staining for Wnt8 in *Xenopus*. Wnt8 is one of the earliest expressed Wnts and important for AP patterning (Kiecker and Niehrs, 2001). Indeed, they found a graded localisation of Wnt8 protein. Furthermore, they show that Wnt8 is localised in clusters at the membrane. These clusters co-localise with heparin sulphate (HS). In the continued analysis, Mii et al. shows that these sugar clusters are sulphated HS clusters. Interestingly, the pre-form of sulphated HS – the acetylated HS – does not co-localise with Wnt8. Sulphated HS clusters are frequently internalised together with Wnt8 and Frz8 leading to signalling eventually. Blockage of HS sulphation is impeding cluster formation and thus signalling. The antagonist Frzb takes Wnt out of the sulphated cluster and translocate Wnt to the acetylated clusters to reduce signalling activity. The authors claim to identified a novel “signalling platform” for Wnt which requires sulphated HS.

The concept put forward to define sulphated HS as a component of the Wnt signalling platform is intriguing. The strong point about this paper is clearly the analysis of the sulphated HS platform and the function of the antagonist Frzb. However, there are several experiments and controls missing to draw these conclusions. First of all it will require more work to convince the readers of the specificity of the used antibodies. Second quantification of the imaging data is hidden in supplementary data or missing at all. Furthermore, the signalling part is not convincing. The authors should use reporter assays and target gene activation to demonstrated the requirement for sulphation of HS during Wnt signalling. A detailed list for experimental suggestions is following.

Furthermore, there are several overlaps with existing hypothesis and the authors do not show/discuss similarities or differences with these data sets:

Firstly, in 2008, a concept of microdomains and Wnt signalling has been put forward from the Kikuchi lab. Here caveolae/cholesterol-positive microdomains has been suggested to be important signalling platforms for beta-catenin dependent signalling. Are these the similar domains? Does sulphated HS co-localise with Caveolae whereas acetylated HS co-localises with Clathrin? The authors should clarify this point and discuss their findings in the context of previously published data.

Secondly, work from the Niehrs, Clevers, He and the Bienz labs have described the architecture of the Wnt signalosome at the plasma membrane in detail. The authors should make use of this findings and proof that sulphated HS is an important new component here, whereas acetylated HS is not.

Thirdly, HSPGs and glypicans have been discussed as co-receptors for Wnt signalling in flies (Carvalho 2010, Sakane, 2012) and vertebrates. *Drosophila* Dally and Dally like have been proposed to function in the beta-catenin signalling pathway, whereas, HSPGs/Glypicans in vertebrates have been implicated in non-canonical Wnt signalling (Kikuchi 2015). How does this fit with the presented data?

Overall, this is an interesting manuscript that describes a novel mechanism in the Wnt signalling platform. However, while many of the experimental approaches are straightforward and support their conclusions more or less directly, other conclusions and their model seem somewhat overreaching. Prior publication there are also several points, detailed below, that need experimental improvements.

In the present form, I regret to say this paper does not meet with the high standard of Nature Communications. Major revisions are required for further consideration.

Detailed review

Fig. 1

The authors should provide further evidence for the specificity of the antibody/staining. One possibility would be to block secretion of Wnt8 (by treatment of the embryos with a porcupine inhibitor) and compare it to the staining in a,b.

Fig1h requires quantification.

Suppl 1

a, only refers to an overexpression scenario –

b, similar in here. Why not use the endogenous Wnt8?

c, Morphants show often down-regulation of protein expression. Therefore it would be more convincing to show staining of another Wnt member in the morphants.

Figure 2

a,b quantification is needed.

d in general the differences are difficult to see as the basic expression levels differ. Furthermore, overexpression of *ndst1* mRNA is missing.

e, staining is not convincing. Non-*ndst1*-injected cells show also a strong beta-catenin localisation in the nucleus. Therefore, a quantification is missing. Could this experiment be repeated with Wnt8 MO? Is there a possibility to proof that sulphated HS is required for signalling in a more direct way?

Supplementary figure 2

b, this is a convincing staining. The author should discuss the similarities to the Huang & Niehrs, 2014.

Supplementary figure 3

f, the authors use STED microscopy to measure the cluster size and find that the clusters do not differ. I find this experiment less interesting. However, one could use this microscopy technique to elucidate the colocalisation of HS with Wnt8.

Figure 4

The authors observe internalisation of Wnt8 within seconds. this is very similar to studies in Hagemann 2014. In which compartments is Wnt8 internalised? Are these early endosomes, late endosomes, recycling endosomes?

Are these caveolae positive of Clathrin positive vesicles? Would this fit with the Kikuchi model?

d, the authors change from Wnt8 to Fzd8 here. Therefore, the authors should provide evidence that Fzd8 is internalized together with Wnt8. Furthermore, it is unclear if Fzd8 co-localise similarly with sulphated HS but not with acetylated HS. Do the authors observe continuous endocytosis of Fzd or is this Wnt triggered?

Figure5 + Suppl figure 10

The co-localisation data are presented in an excellent way – unfortunately partially hidden in the

sup fig 10. The authors should put these findings in the centre of the paper. Co-localisation data should be backed-up with biochemical methods such as co-ip experiments. Additional, experiments are required to elucidate the molecular nature of the signalling / non-signalling platforms, a link to endocytosis is required and the consequences on signalling has to be shown more in detail

Response to reviewers' comments

We greatly appreciate all of you for providing us with a lot of helpful comments.

We now answer all of the points raised by the reviewers, shown hereinafter. Especially, we present the result of a biochemical analysis, including electrophoretic separation of HS chains followed by immunodetection with NAH46 and HepSS-1 antibodies (CAME-AIS analysis). With this analysis, we would answer the substantial criticism pointed out by the reviewer #1 and the editor regarding characterization of the HS chains recognized with the antibodies. As a result, we can discuss more clearly and precisely the nature of the HS chain recognized by these antibodies.

We believe that this manuscript has been greatly improved as a result of these changes. Below, the reviewer's comments are pasted in blue, followed by our replies in black.

Comments from the Reviewer #1:

This interesting manuscript addresses the regulation of Wnt8 in *Xenopus*, with particular emphasis on the roles of heparan sulfate proteoglycans, and exogenous inhibitor, Frzb. The authors utilize two HS-specific antibodies, both IgMs, to propose that there are discrete populations of HS on cell surfaces, one sulphate-rich the other sulphate poor, corresponding with the known specificities of the antibodies. They present data to suggest that sulphate-rich pools, on glypicans, have internalization/signaling potential, while the sulphate –poor pool, also on a glypican, does not. Moreover, Frzb, a possible soluble inhibitor of Wnt localizes to the sulphate-poor HS clusters and sequesters Wnt8 and thereby diminishes signaling.

These are interesting concepts, supported by high quality micrographs, and there is much of interest to specialists and perhaps others, but it appears that in many places the data are over-interpreted. Some areas of concern are as follows:

1. The authors contend that they have identified an entirely new paradigm for HS structures, and propose that the sulphate-poor clusters represent unsulfated heparosan. This is distinct from the high sulphated clusters. While the authors may have shown that HS with different sulfation patterns exist in a single tissue (a potentially novel and interesting finding), that is as far as it goes without characterizing the chains. The NAH46 antibody was raised against K5 polysaccharide (which is unsulfated), but the fact that it

recognizes vertebrate HS does not show the presence of totally unsulfated chains. Possibly the unsulfated (NA) domains are larger or more numerous than in other HS pools. Moreover, the authors' own data shows that the high-sulfated pool, recognized by the HepSS-1 antibody, also contains NA domains. This is because the authors show sensitivity of the high-sulfated pool to heparitinase, an enzyme that only hydrolyzes NA domains. The authors also cannot rule out that the selective ability of the two antibodies to bind different HS pools is simply steric, since both are large IgM molecules. Short of purifying and characterizing the chains, the authors cannot justify their claim of an entirely new concept in HS synthesis, particularly when there is so much evidence to the contrary in the literature. Chains of varying fine structure are well known, that may be all that is discriminated here.

Ans: First of all, we thank the reviewer #1 for the helpful comments and suggestions. We agreed with this comment regarding characterization of the HS chains and considered that our previous data were not sufficient to support the model presented in the previous manuscript. Therefore, we performed biochemical analyses to examine HS chains recognized by NAH46 and HepSS-1 antibodies in collaboration with Drs. Shuhei Yamada and Shuji Mizumoto, who are specialists in glycosaminoglycan biology.

We first attempted to examine HS chains in *Xenopus* gastrula by co-IP experiments using NAH46 and HepSS-1 followed by a standard disaccharide analysis. However, we did not recover any detectable amount of immunoprecipitated HS chains, probably due to the nature of IgMs, which are in general not applicable for co-IP.

As an alternative approach, we next examined glycosaminoglycans (GAGs) purified from *Xenopus* embryos using "cellulose acetate membrane electrophoresis (CAME)." In CAME, substances are separated based only on their net negative charge density, without regard to size (Fig. 2f). To detect HS chains on the membrane using antibodies, we developed "alcian immunostaining" system (named CAME-AIS, Fig. 2g), a similar procedure to western blotting after SDS-PAGE. Using this method, we successfully showed that HS chains are differentially recognized by NAH46 and HepSS-1 (Fig. 2g, magenta). HepSS-1 signal was mainly detected in a relatively lower area (relatively negative, Area 4), and only weakly in a relatively upper area (Area 3). In contrast, strong NAH46 signal was detected in Area 3 (Fig. 2g, short exposure), and relatively weak but evident signal was also detected in Areas 2 and 4 as well (Fig. 2g, long exposure). These data suggest that populations of HS chains recognized by these two antibodies are not identical, but are in fact biochemically separable at least in part. On the other hand, the wide distribution of HS detected by NAH46 suggests that the relative content of each

modification of a single GAG chain is variable and that even N-sulfo-rich HS appears to contain some consecutive N-acetyl disaccharide units (epitopes of NAH46) (Fig. 2g).

To examine the relevance of this *in vitro* study to the cytological observations, we carefully reconsidered our previous immunostaining experiments with NAH46 and HepSS-1 antibodies in *Xenopus* embryos (Fig. 2a-c). As already described in the previous text, the lack of a strong correlation between NAH46-positive puncta (N-acetyl-rich) and HepSS-1-positive ones (N-sulfo-rich) along the cell boundary indicated that N-acetyl-rich and N-sulfo-rich HSs are individually clustered on the cell membrane (Fig. 2c). On the other hand, consistent with the biochemical analysis showing that even N-sulfo-rich HS contains N-acetyl disaccharide units, quantification of double staining signals along the cell boundary also showed that almost all HepSS-1 puncta were more or less detected with NAH46 signals (Text p. 10, lines 166-169; Fig. 2b). With these new findings, we can now understand that the distinct clustering of N-acetyl-rich HS and N-sulfo-rich HS (Supplementary Fig. 1c), which we originally report in this paper, does not conflict with the general consensus of the characteristics of HS chains, which contain both NA and NS domains within an individual chain (NA/NS domain model; Supplementary Fig. 1b). We added description about these points in the text (p. 13-14, lines 232-243; p. 26, lines 455-466).

In addition, we would like to show our replies for several points raised by this reviewer.

1) About the specificity of heparitinase activity

We used “heparitinase” purchased from Seikagaku Corp. (cat # 100703) in the enzymatic digestion assays (Fig. 1d, Supplementary Fig. 4c). According to information from this manufacturer, this product is a mixture of heparitinase I and heparitinase II at 4:1 ratio. We should have made it clear in the previous text and we apologize for it. As shown by Nader et al., (1990) in *J. Biol. Chem* and Nadar et al., (1999) in *Glycoconj. J.*, heparitinase I acts on both N-acetylated and N-sulfated glucosaminido-glucuronic acid linkages of the heparan sulfate. Therefore, it is plausible that the heparitinase used in this study cleave epitopes of HepSS-1 as well as those of NAH46. We added description about the activity of the heparitinase in the text (p. 7-8, lines 125-126).

2) About steric problem of NAH46 and HepSS-1 IgM molecules

We also considered possible steric effect of IgM in recognition of epitopes. First, steric interference between NAH46 and HepSS-1 was not likely to occur in our double staining experiments (Fig. 2a), because the staining patterns of these antibody were basically the

same as single staining with NAH46 or HepSS-1 (Fig. 1e). Furthermore, in double staining, overlapped signals of NAH46 and HepSS-1 were observed at some puncta (Fig. 2a, white arrowheads), suggesting that these IgM antibodies do not have severe steric interference with each other.

2. The authors make much of “internalized” Wnt, but it is very unclear how they know this. The methods are not sufficiently detailed to be sure. Puncta that are not at the cell margins (cell-cell interaction sites) are not necessarily internal. In addition, “signalosomes” are identified through Dvl2/Fzd8 codistribution. However, staining for Dvl2 is very sparse, and given the extensive Fzd8 staining in many cases, specificity of the codistribution patterns is not entirely convincing. If the authors wish to prove interaction here and elsewhere, then well-known FRET methods would have to be used. There are similar concerns with the beta-catenin data, nuclear staining, where present, is far from uniform and generally unconvincing.

Ans:

1) How we recognized the internalization of Wnt.

As this reviewer pointed out, our description about the internalization of Wnt was not sufficient in the previous manuscript. Thus, we show further explanation on this issue as follows.

In the case of overexpression of mV-Wnt8 or Wnt8-Myc, Wnt8-expressing cells are strongly stained including the intracellular region, which is easily distinguishable from surrounding non-expressing cells. If such non-expressing cells have punctate staining of Wnt8 inside the cells, those Wnt8 proteins must be internalized from the outside because we observed the subapical region of the cells (basal to the tight junction, not the apical cell surface), by confocal microscope. In the case of the endogenous Wnt8, *wnt8* gene is expressed in the mesoderm, but not in the superficial layer, where we observed the distribution of Wnt8. Therefore, if puncta of Wnt8 are detected inside the superficial cells, they must be internalized.

In addition, we had paid attention to the depth (z-direction) of the optical section (xy plane) in our observations with confocal microscopy (see also Supplementary Fig. 12i). As shown in Supplementary Fig. 3b for Wnt8-Myc, secreted Wnt8 as well as Frzb was detected in the basolateral intercellular space. Thus, we can distinguish internalized proteins from ones on the cell surface. We added explanation for how we regard intracellular Wnt staining as internalized one (p. 19, lines 332-338).

2) Identification of signalosome components with N-sulfo HS

We agreed to this reviewer's claim that the staining of Dvl2 itself was not very convincing although overlapping of Dvl2 and Fzd8 was enhanced by Wnt8. To solve this problem, we examined phosphorylated Lrp6, which is generally recognized as a good indicator of the Wnt signalosome (Bilic et al, 2007, *Science*), by using an anti-phospho-LRP6 (pLRP6) antibody. Fortunately, we successfully obtained more reliable results with this antibody than with anti-Dvl2 antibody (Fig. 7). We found that the phosphorylation of LRP6 actually occurred depending on Wnt8 in the *Xenopus* embryo (Supplementary Fig. 10), and showed that phosphorylated LRP6 is highly colocalized with N-sulfo HS but not with N-acetyl HS upon Wnt8 expression (Fig. 7a-f). Furthermore, the staining of phosphorylated LRP6 was reduced in *ndst1* morphants, which was rescued by *ndst1* mRNA (Fig. 7h). To further strengthen this result, we performed immunostaining of phosphorylated LRP6 and N-sulfo HS in HeLa cells, which is used in the first report of the Frizzled/Wnt/LRP6 signalosome by Bilic et al (2007). We also showed that phosphorylated LRP6 and N-sulfo HS were highly colocalized in HeLa cells (Supplementary Fig. 13f,g). Considering these new data, we revised the text (p. 20-22, lines 345-393).

In contrast, in spite of the suggestion by this reviewer we wonder that FRET is not applicable for this purpose. As far as we understand, it is generally difficult to apply FRET for evaluating interaction between two independent molecules because the molecular ratio of CFP and YFP could not be precisely manipulated. Most of the FRET applications use a single molecule probe such as Ca²⁺ probe cameleon (Miyawaki et al., 1999).

3) beta-catenin staining

Following to the reviewer's suggestion, we newly performed knockdown experiments for *ndst1* (Fig. 5g,h), and analysed the nuclear localization of beta-catenin. In this experiment, a low Wnt-activity area was observed for *ndst1* overexpression, whereas a high Wnt-activity area was observed for *ndst1* knockdown. Such description about the analysis is not sufficient in the previous text and figures. We apologize for it and added explanations and a new schematic figure for the experimental procedure (Fig. 5d). Furthermore, since beta-catenin staining in the nucleus is not so uniform in general when observed by confocal microscopy, we counted the number of cells with nuclear staining of beta-catenin as judged by overlapping with DAPI nuclear staining, and performed statistical analysis (Fig. 5f,h). These data suggest that *ndst1* is necessary and sufficient

for the activation Wnt signalling.

3. An important starting point for this data is the codistribution of endogenous Wnt8 with high-sulfated HS (Fig 1H for example). However, a very significant portion of the Wnt8 is not apparently colocalized with the HS pattern. If the Wnt8 is lost by NDST1 MO (Fig 2D but needs careful quantitation), then there are quite possibly HS pools not detected by the HepSS-1 antibody. The authors seem to not take this possibility into account.

Ans: As this reviewer pointed out, our data in the previous version did not clearly show codistribution of the endogenous Wnt8 with high-sulfated HS. To improve this point, we carefully repeated this experiment again and finally obtained a solid data showing most Wnt8 puncta are colocalized with HepSS-1 staining (Fig. 1f,i), probably due to improvement of Wnt8 staining. Furthermore, we quantified stained signals for Wnt8 and N-acetyl-rich or N-sulfo-rich HS along the cell-cell border in the newly obtained images (Fig. 1g,j), and calculated correlation coefficient values (Fig. 1h,k) for multiple samples with a statistical analysis (p. 8, lines 139-142). With this analysis, we showed that the correlation coefficient value between Wnt8 and N-sulfo-rich HS is significantly higher than that between Wnt8 and N-acetyl-rich HS, suggesting that endogenous Wnt8 associates with N-sulfo-rich HS (p. 8, lines 134-143).

On the other hand, there still remains some parts of the Wnt8 was not colocalized with HepSS-1 staining (Supplementary Fig. 12g, open arrowheads). This difference between Fig. 1i and Supplementary Fig. 12g might reflect different developmental stages (st. 10.5 and 11.5, respectively). As the reviewer pointed out, some other HS pools that are not detected by HepSS-1 might colocalize with Wnt8. One possible explanation is that HepSS-1 does not react with some structures of HS that are modified by epimerization and O-sulfation (van den Born et al., 2005, *J. Biol. Chem.*), and these modified HS could also be involved in Wnt distribution and signalling. Indeed, analysis of sulf-1, which hydrolyzes 6-O-sulfation of HS, indicates that 6-O-sulfation enhances binding of Wnt to HS but down-regulates Wnt signalling (Ai et al, 2003, *J. Cell Biol.*). We have discussed this in the text (p. 30-31; lines 533-539).

4. The authors use fluorescence microscopy almost entirely, but this mostly lacks any quantitation and there are places where biochemical techniques should be used to confirm the imaging data. What affinities does Wnt8 have for HS, does it require high-sulfated (NS domains) of HS? Does Frzb have high affinity for NA domains of HS? It is difficult to imagine that Wnt8 can transfer from NS-rich HS to NS-poor HS since the former will

almost certainly have higher affinity. Presumably Frzb competes for the Wnt8, which is itself enriched at sites of high NA content (nicely shown and potentially important but see point 5) and this mediates the transfer of Wnt8?

Ans: According to the reviewer's suggestion, we quantified a number of images that were crucial in this manuscript (revised Figures 1f-k, 2d,e, 5b,c,e-h, 7a-f, Supplementary Fig. 12a-f; and for HeLa cells, Supplementary Fig. 13).

We also attempted to obtain the biochemical evidence supporting our microscopic observations. We performed co-immunoprecipitation (co-IP) of HS, and Wnt8 or HA-tagged-Frzb overexpressed in *Xenopus* embryos using NAH46 or HepSS-1 antibody and anti-Wnt8 or HA antibody, followed by western blotting. However, as mentioned above, these experiments were technically too difficult to obtain reliable results.

In spite of the lack of biochemical analyses, we believe that our results obtained by increasing and decreasing *ndst1* function clearly indicate that Wnt8 binding requires NS domains of HS and Frzb has high affinity for NA domains of HS (revised Figures 5a-c, 9a, Supplementary Fig. 14c).

With regard to Wnt transport from NS to NA clusters, we speculate a similar mechanism to that proposed by this reviewer; that is, this transfer is mediated by competition by Frzb. This speculation is supported by the following two data: (i) Wnt8-Myc, which is colocalized with N-sulfo-rich HS in the absence of Frzb, changed to colocalize with N-acetyl-rich HS when Frzb is expressed in neighboring cells (Fig. 8), suggesting that Wnt8 colocalized with N-sulfo-rich HS turned to associate with N-acetyl-rich HS through interaction with Frzb in the extracellular space; (ii) Accumulation of mV-Wnt8 on the surface by *ndst1*-overexpression could be decreased by coexpression with Frzb (Fig. 9a, this is a new data.). Although we have not observed dynamic processes of transferring Wnt8 from NS-rich HS to NA-rich HS by Frzb, our new data suggest that Frzb can apparently switch colocalization of Wnt8 from N-sulfo-rich HS to N-acetyl-rich HS (Figs. 8c and 9a). We have incorporated such discussion in the text (p. 23-25, lines 412-436).

5. Figure 5. Why is the staining pattern for N-sulfo HS so different here compared, for example, with Fig 2? Is it likely that over-expression of Frzb disturbs the distribution of N-sulfo HS and indeed may be lost from the cell surface? If so, this introduces a potential problem of interpretation. The idea that Frzb localizes to a non-signaling compartment of HS is intriguing and very interesting, but there needs to be caution applied here to data interpretation.

Ans: We understand that the critical question raised by this reviewer is whether Frzb affects staining patterns of HepSS-1. Through our careful observations of the HS staining, we noticed that staining pattern of N-sulfo HS by HepSS-1 appeared to be more variable among cells comparing with that of N-acetyl HS by NAH46. The mechanism underlying the variation of HepSS-1 staining is currently unknown, however Kure and Yoshie (1986) have already described a similar observation even within a single cell line (NIH3T3), possibly reflecting cell density and/or cell growth. We added this description in p. 9, lines 154-157 and p. 73, lines 1376-1379 (Supplementary Fig. 4a,b). However, as shown in our new Fig. 8, overexpression of Frzb neither disturbs the distribution of N-sulfo HS nor is lost from the cell surface.

6. The STED microscopy, while a current technique, seems not to add much. The authors note a difference in dimensions between the sulphate-rich and sulphate-poor HS clusters. However, they should perhaps note that this difference is in the order of the diameter of an IgM antibody (35-45nm) and so may not be informative.

Ans: Thank you for this comment. We agree to this possibility, and hence added the following statement in the legend (p. 74-75, lines 1399 to 1400): “It should be noted that IgM is about 19 nm in a diameter (*PNAS* 106:14960, 2009), possibly causing overestimation of the sizes of two types of HS clusters.” In addition, we removed statistical analysis for the difference of the sizes, because the difference of the estimated size between the two might be just reflected by the difference of IgM sizes between NAH46 and HepSS-1.

7. Supplementary Fig 8. Clearly much of the over-expressed Wnt8 and Fzd8 does not colocalize with the N-sulfo clusters. Moreover it seems clear, by comparison of Figs a and b, that over-expression of Wnt8 alters the nature of the Fzd staining considerably. This does not negate the very nice work on endogenous Wnt8 but shows that caution should be applied to data interpretation.

Ans: We are very sorry that some of the panels did not show good images. We repeated the same experiments with improved immunostaining, and obtained better pictures instead of previous Supplementary Fig. 8 (now Supplementary Fig. 12g). We presented clear images for N-sulfo HS, Fzd8 and their overlap as well as Wnt8. On the other hand, as discussed above for point 3, some portions of Wnt8 were not colocalized with N-sulfo

HS, probably due to the developmental stage (st. 11.5) (Supplementary Fig. 12g). Furthermore, we also carefully examined the possibility that overexpression of Wnt8 alters the nature of the Fzd staining. We repeatedly observed that Wnt8 increased the amount of Fzd8 on the cell membrane (Supplementary Fig. 12g, middle and bottom panels). In addition, we also found that Fzd8 staining was reduced by *ndst1* knockdown (Fig. 7g). Since these results suggest that the stabilization or expression of Fzd8 was dependent on Wnt8, we discussed our results (p. 21-22, lines 365-373; 376-382).

8. A minor point. In the very useful experiments to deplete either GPC4 or 5, does depletion of one affect the expression of the other? This may have bearing on the results.

Ans: We consider that the possible effects suggested by this reviewer were not dramatic to affect our results. As this reviewer suggested, knockdown (KD) of *gpc4* or 5 might affect the expression of the other for compensation. If this is the case, our overexpression data can predict following results. If *gpc4* KO enhances *gpc5* expression, it increases the level of N-sulfo HS. In contrast, if *gpc5* KO enhances *gpc4* expression, it relatively reduces N-sulfo HS. However, we did not observe such responses in the KD embryos.

Comments from the Reviewer #2:

This study aims to understand the distribution of Wnt signaling proteins in vivo, in *Xenopus* embryos. Using a newly generated antibody to Wnt8, the authors show the graded distribution of the protein to membrane puncta and their partial colocalization with clustered heparan sulfate proteoglycans (HSPGs). The Wnt8 protein was only associated with the N-sulfo HSPG molecules, one of the two major modifications of HSPG chains. Finally, the authors demonstrate that the N-sulfo clusters are frequently internalized and play a positive role in Wnt signaling. By contrast, N-acetyl HSPG accumulate Frzb, a secreted Wnt antagonist. The authors further show that glypican 5 is a major core protein for the N-sulfo clusters, whereas glypican 4 contributes to both N-sulfo and N-acetyl clusters. Based on the colocalization of the Wnt8-N-sulfo clusters with Frizzled and Dishevelled, the authors propose that the HSPG clusters represent predefined sites for signalosome formation.

Overall, the reported findings reflect a significant effort, are highly original and would be

of interest to the field. The visualization of N-acetyl- and N-sulfated (N-sulfo) modifications of HSPGs as separate clusters *in vivo* has never been reported. Similarly, a role of HSPGs in Wnt signaling has been known, but the association of a Wnt protein with a specific form of HSPG is a significant advance corroborated in this study by the knockdown of the relevant NSDT enzyme and glypicans. The majority of the experiments include relevant controls for antibody and morpholino specificity. There is no doubt that these findings should be published, however, the remaining question is whether the authors' conclusions are sufficiently supported by data. The authors are encouraged to take into account the following two comments.

We thank reviewer #2 for his or her helpful comments and criticisms.

1. The work would be strengthened by biochemical evidence for the existence of separable 'N-sulfo' and 'N-acetyl' clusters and the Wnt8-N-sulfo complexes. This could be done in pull down assays with the same antibodies that were used for immunohistochemistry.

Ans: In accordance with reviewer 2's suggestion, we have tried immunoprecipitation using the NAH46 (for N-acetyl HS) and HepSS-1 (for N-sulfo HS) antibodies and protein L beads, but failed to obtain specific precipitant(s), probably because the antibodies we used are IgM, which have been postulated to be unsuitable for immunoprecipitation due to insufficient affinities for epitopes. We have also tried co-IP of embryonic lysates containing overexpressed Wnt8 or Frzb-HAi using anti-Wnt8 or anti-HA antibody, respectively, followed by Western blotting analysis with NAH46 and HepSS-1. However, we could not detect specific bands with NAH46 and HepSS-1, at least partly due to insufficient affinity of IgM. So, we took another biochemical approach, in which glycosaminoglycans purified from *Xenopus* embryos were analyzed by a newly developed method, named "cellulose acetate membrane electrophoresis coupled with alcian immunostaining (CAME-AIS)," an immunodetection system of glycosaminoglycans immobilized on the cellulose acetate membrane with alcian blue after membrane electrophoresis (Fig. 2g). As a result, we have successfully shown that glycosaminoglycans from *Xenopus* embryos were at least partly separated into "N-sulfo-rich" and "N-acetyl-rich" HS in the electrophoresis (p. 13, lines 222-232).

To examine *in vitro* complex formation of Wnt8 and N-sulfo HS or that of Frzb and N-acetyl HS, we used embryonic lysates containing overexpressed Wnt8 or Frzb-HAi and tried their binding to *Xenopus* glycosaminoglycans immobilized on the cellulose acetate

membrane after membrane electrophoresis, similar to CAME-AIS. However, we could not obtain a clear result, maybe due to sensitivity of this method.

2. The proposed role of N-sulpho HS in Wnt signalosome formation can be further confirmed by monitoring the distribution of Wnt8 protein added to HeLa cells. This is an independent system, in which the N-sulpho and N-acetyl HS clusters are differentially distributed (this study) and the signalosomes were first characterized (Bilic et al., 2007). Signalosome formation can be corroborated by the appearance of a specific modification, such as LRP6 phosphorylation.

Ans: In accordance with the reviewer's suggestion, we examined the distribution of N-sulfo and N-acetyl HS clusters with signalosome components using HeLa cells. As shown in new Supplementary Fig. 13 (p. 20, lines 356-359), N-sulfo HS is preferentially colocalized with phosphorylated LRP6 (pLRP6), compared to N-acetyl HS, in HeLa cells. Similarly, we showed that pLRP6 was highly colocalized with N-sulfo HS but not with N-acetyl HS in *Xenopus* embryos (Fig. 7a-f). Furthermore, *ndst1* knockdown experiments showed the requirement of N-sulfation for the phosphorylation of LRP6 (Fig. 7h).

Other points

--The signaling assays are based on embryonic phenotypes and beta-catenin localization, but proper quantification is lacking.

--Does overexpressed *Ndst1* (alone or together with Wnt8) increase TOP-Flash activity?

Ans: Following the reviewer's suggestion, we quantified nuclear staining of beta-catenin in *ndst1*-overexpressed cells and newly added a knockdown experiment (Fig. 5d-h), showing statistically significant differences with *ndst1* overexpression or knockdown as expected.

In addition, we carried out TOP-Flash assays. In Wnt8-overexpressed conditions, we examined the effect of *ndst1* overexpression as well as knockdown, on Wnt signaling (Fig. 5i-k). Our results indicated that TOP-Flash activity is upregulated by *ndst1* overexpression (Fig. 5i), and downregulated by *ndst1* MO (Fig. 5k). Furthermore, this MO effect was rescued by a small amount of *ndst1* mRNA (Fig. 5i,k).

-- Fig. 1H, Fig.4C, Fig S3E, Fig S8a. The conclusions are based on puncta colocalization, which is sometimes difficult to see.

Ans: We apologize for unclear pictures. We improved these pictures as shown hereinafter:

For Fig. 1h, we did immunostaining again and quantified the staining intensities. Now the correlation between distributions of endogenous Wnt8 and N-sulfo HS was significantly higher than that between Wnt8 and N-acetyl HS (new Fig. 1f-k).

For Fig. 4c, we adjusted contrast of the pictures (new Fig. 4b).

For Fig. S3e, we added arrowheads and outlines of the cells (new Supplementary Fig. 4g).

For Fig. S8a (Fzd8 staining was somewhat cloudy in the previous figure), we improved immunostaining and replaced it with new pictures (new Supplementary Fig. 12g). In the previous version, we showed the overlap of immunostained signals for N-sulfo HS, Fzd8, and overexpressed Wnt8. For the revised version, we compared the overlap between N-sulfo HS and Fzd8 with or without overexpressed Wnt8 in the mDMZ region. The data suggest that Fzd8 colocalizes with N-sulfo HS upon overexpression of Wnt8.

--Fig S4B. To exclude a nonspecific effect of the NDST1 morpholino on RNA translation, Wnt8 levels can be shown.

Ans: For the secondary axis assay (new Supplementary Fig. 9b), we injected a small amount (5 pg/embryo) of *wnt8* mRNA, which is lower than the sensitivity of immunostaining as well as western blot (please also note that the secondary axis induction with injected *wnt8* mRNA should occur during early cleavage stages). Regarding a nonspecific effect of MO, we have already done rescue experiments with a small amount of *ndst1* mRNA, which showed partial recovery of secondary axis formation. Fig. 2e also shows that *ndst1* MO rather increases N-acetyl staining, suggesting that *ndst1* MO does not exhibit nonspecific effects on global translation.

--Is the phenotype of the *Ndst1*-knockout mice (Ringvall et al., 2000, JBC; Grobe et al., 2005; Development) consistent with the proposed role of this enzyme in Wnt signaling?

Ans: In the *Ndst1*-KO studies by Ringvall et al. and Grobe et al., Wnt signalling was not examined. However, in their following paper by Pallerla et al. (2007), the authors described that “30% of the mutants show reduced canonical Wnt signaling in the presumptive midbrain, hindbrain, and spinal cord.” This reduction of canonical Wnt signalling is consistent with our proposed role of *Ndst1* enzyme in Wnt signalling. We cited their work and revised the text accordingly (p. 4, line 67).

Comments from the Reviewer #3:

The Wnt signalling pathway is one of the most important signalling pathways during development and tissue homeostasis. It regulates cell proliferation, stem cell maintenance, cell differentiation and regeneration of various organs. During development, Wnt/beta-catenin signalling is important organizing the embryonic body by setting the anterioposterior. Therefore, Wnt falls into the class of form-giving substances – so-called morphogens. To fulfil its morphogenetic function it is released from a localised source and forms a long-range signalling gradient. However, the endogenous Wnt protein gradient has not been shown in vertebrates so far. Mii and colleagues start with introducing an immune-staining for Wnt8 in *Xenopus*. Wnt8 is one of the earliest expressed Wnts and important for AP patterning (Kiecker and Niehrs, 2001). Indeed, they found a graded localisation of Wnt8 protein. Furthermore, they show that Wnt8 is localised in clusters at the membrane. These clusters co-localise with heparin sulphate (HS). In the continued analysis, Mii et al. shows that these sugar clusters are sulphated HS clusters. Interestingly, the pre-form of sulphated HS – the acetylated HS – does not co-localise with Wnt8. Sulphated HS clusters are frequently internalised together with Wnt8 and Frz8 leading to signalling eventually. Blockage of HS sulphation is impeding cluster formation and thus signalling. The antagonist Frzb takes Wnt out of the sulphated cluster and translocate Wnt to the acetylated clusters to reduce signalling activity. The authors claim to identified a novel “signalling platform” for Wnt which requires sulphated HS.

The concept put forward to define sulphated HS as a component of the Wnt signalling platform is intriguing. The strong point about this paper is clearly the analysis of the sulphated HS platform and the function of the antagonist Frzb. However, there are several experiments and controls missing to draw these conclusions. First of all it will require more work to convince the readers of the specificity of the used antibodies. Second quantification of the imaging data is hidden in supplementary data or missing at all. Furthermore, the signalling part is not convincing. The authors should use reporter assays and target gene activation to demonstrated the requirement for sulphation of HS during Wnt signalling. A detailed list for experimental suggestions is following.

Furthermore, there are several overlaps with existing hypothesis and the authors do not

show/discuss similarities or differences with these data sets:

Firstly, in 2008, a concept of microdomains and Wnt signalling has been put forward from the Kikuchi lab. Here caveolae/cholesterol-positive microdomains has been suggested to be important signalling platforms for beta-catenin dependent signalling. Are these the similar domains? Does sulphated HS co-localise with Caveolae whereas acetylated HS co-localises with Clathrin? The authors should clarify this point and discuss their findings in the context of previously published data.

Ans: We thank the reviewer #3 for helpful comments and suggestions. We performed immunostaining of caveolin and N-sulfo/N-acetyl HS in HeLa cells (Supplementary Fig. 13h,i). However, this staining was difficult in *Xenopus* embryos due to cross-reactivity of the antibody. In HeLa cells, we observed that caveolin was well colocalized with N-sulfo HS, not with N-acetyl HS even in a condition without exogenous Wnt, consistent with our data about internalization of HS (Fig. 4) as well as the report from Kikuchi lab that showed the requirement of caveolin for the internalization of LRP6 and canonical Wnt signalling (Yamamoto et al., 2006, *Dev. Cell*) (Supplementary Fig. 13h-i). We described this point in the revised text (p. 22, lines 386-389). Unfortunately, we could not examine the colocalization of clathrin with the two types of HS because all of the antibodies available for immunostaining are generated by mice. However, we speculate that N-acetyl HS clusters are colocalized neither with caveolin nor clathrin by the following reasons. (1) Although caveolin and clathrin are used for the internalization of proteins from the plasma membrane, N-acetyl-rich HS clusters are rarely internalized (Fig. 4). (2) Both types of HS cluster are based on glypicans, and it was suggested that they are in the lipid rafts (Supplementary Fig. 7b). Clathrin is reported to be involved in the endocytosis of non-lipid rafts. These data suggesting that N-acetyl HS is very likely to stay in lipid rafts without association to caveolin or clathrin.

Secondly, work from the Niehrs, Clevers, He and the Bienz labs have described the architecture of the Wnt signalosome at the plasma membrane in detail. The authors should make use of this findings and proof that sulphated HS is an important new component here, whereas acetylated HS is not.

Ans: In accordance with the reviewer's suggestion, we examined the distribution of N-sulfo-rich and N-acetyl-rich HS clusters with phosphorylated LRP6 (pLRP6), which is a direct and crucial step in the Wnt signal transduction as well as the major component of the Wnt-LRP6 signalosome, using *Xenopus* embryos and HeLa cells. In *Xenopus*

embryos, we revealed that pLRP6 is highly colocalized with N-sulfo HS but not with N-acetyl HS along the cell boundary (Fig. 7a-f). Furthermore, knockdown of *ndst1* reduced pLRP6 staining, suggesting that N-sulfo HS is required for phosphorylation of LRP6 (Fig. 7h). Using HeLa cells, we also revealed that N-sulfo HS but not N-acetyl HS, is highly colocalized with pLRP6 (Supplementary Fig. 13f,g).

Thirdly, HSPGs and glypicans have been discussed as co-receptors for Wnt signalling in flies (Carvalho 2010, Sakane, 2012) and vertebrates. *Drosophila* Dally and Dally like have been proposed to function in the beta-catenin signalling pathway, whereas, HSPGs/Glypicans in vertebrates have been implicated in non-canonical Wnt signalling (Kikuchi 2015). How does this fit with the presented data?

Ans: In *Drosophila*, Dally protein is important for Wnt signal activation and Dally-like protein is important for the distribution of Wnt. Gpc5 (also Gpc3) belongs to the Dally family, whereas Gpc4 (also Gpc1, Gpc2, and Gpc6) belongs to the Dally-like family. We have shown that Gpc5 is one of the core proteins of N-sulfo-rich HS (involved in Wnt/beta-catenin signalling), whereas Gpc4 is the core protein of N-acetyl-rich HS (involved in Wnt distribution with Frzb). These results are consistent well with the *Drosophila* data as mentioned in Discussion (p. 29, lines 504-519). In addition, our data showed that Gpc4 is the core protein for both N-sulfo-rich and N-acetyl-rich HS whereas Gpc5 is only for N-sulfo-rich HS. Therefore, knockdown of Gpc4 only affects N-acetyl-rich HS, but not N-sulfo-rich HS due to functional redundancy with Gpc5. This may explain why “HSPGs/Glypicans (Gpc4) in vertebrates have been implicated in non-canonical Wnt signalling” as has been shown by Ueno’s group (Development 130:2129-38, 2003). Although we could not find a paper from Kikuchi et al. in 2015 that the reviewer suggested, in the paper from Kikuchi’s lab (Sakane et al. (2012)), they showed that knockdown of Gpc4 suppresses both canonical and noncanonical pathways. Although we cited this paper in the previous text, we did not mention this result. Thus, we described their results in this revised version. In addition, we also cited a paper by Capurro et al. (2014), showing that Gpc3 (Dally family) is involved in canonical Wnt signalling. We discussed these points in the revised text (p. 28-29, lines 497-504).

Capurro, M., Martin, T., Shi, W. and Filmus, J. (2014). *J. Cell Sci.* 127, 1565-75.

Overall, this is an interesting manuscript that describes a novel mechanism in the Wnt signalling platform. However, while many of the experimental approaches are

straightforward and support their conclusions more or less directly, other conclusions and their model seem somewhat overreaching.

Ans: To avoid overreaching of our statement, we changed our model in Supplementary Fig. 1c and Fig. 9c, d (previous Fig. 5e, f) as follows. In the previous model, N-acetyl HS and N-sulfo HS clusters appear to be fully N-acetylated (meaning “heparan”) and fully sulfonated, respectively. However, with further biochemical experiments (Fig. 2g), we became to recognize that this model is overreaching especially at the composition of HS chains (please read our comments to the first question raised by the reviewer #1). Thus, we changed the model and added more explanation to avoid misleading (p. 26, lines 455-466).

Prior publication there are also several points, detailed below, that need experimental improvements.

In the present form, I regret to say this paper does not meet with the high standard of Nature Communications. Major revisions are required for further consideration.

Detailed review

Fig. 1

The authors should provide further evidence for the specificity of the antibody/staining. One possibility would be to block secretion of Wnt8 (by treatment of the embryos with a porcupine inhibitor) and compare it to the staining in a,b.

Suppl 1 (moved from the below for the same topics)

a, only refers to an overexpression scenario –

b, similar in here. Why not use the endogenous Wnt8?

c, Morphants show often down-regulation of protein expression. Therefore it would be more convincing to show staining of another Wnt member in the morphants.

Ans: As this reviewer suggested, we tried to use a Porcupine inhibitor, IWP-2, for *Xenopus* embryos. We added IWP-2 into a culture medium or injected it into the blastocoel of *Xenopus* embryos (this method can deliver drugs that may not penetrate through the vitelline membrane or the superficial layer of the embryo). However, the treated embryos did not show any phenotypes that are supposed to be caused by Wnt inhibition, such as enlarged heads. This may be due to low solubility of IWP-2 below 22°C, which is a suitable temperature for the development of *Xenopus* embryo. In fact,

we could not find any reliable literature that used IWP-2 for *Xenopus* embryos.

The reviewer requested us to show “the specificity,” but IWP-2 inhibits secretion of all Wnt ligands, which may not tell us the specificity. So, the reviewer might have also asked us “the sensitivity” of antibody detection or “cross-reactivity” for endogenous unknown proteins. In terms of the specificity of anti-Wnt8 antibody, Fig. 1 and Supplementary Fig. 2 have shown the following multiple lines of evidence.

1. Signal was NOT detected with the preimmune serum of the same rabbit as an immunized one (Fig. 1c).
2. Overexpressed Wnt8-Myc, but not Wnt3a-Myc, was detected by wholemount immunostaining of *Xenopus* embryos with anti-Wnt8 antibody, showing the specificity (the antibody does not cross-react with Wnt3a) (Supplementary Fig. 2a). Furthermore, as shown in Fig. 1a and Supplementary Fig. 12g, Wnt8 staining is little or weak in the mid-dorsal region (mDMZ), compared to the lateral or ventral side (LMZ or VMZ, respectively). Thus the mDMZ is a good negative control region for Wnt8 staining, showing the specificity and sensitivity. However, the antibody might have cross-reacted with some unknown proteins that are distributed in lateral to medial (dorsal) gradient manner. To eliminate this possibility, we carried out knockdown of *wnt8*. The data showed that staining with anti-Wnt8 antibody was reduced in the ventral region by injection of *wnt8* MO (Supplementary Fig. 2c). We also examined off-target effects of *wnt8* MO by checking protein production in injected cells as assayed by whole-mount immunostaining with anti-beta-tubulin antibody (not shown).
3. Overexpressed Wnt8-Myc in the embryos was specifically detected by western blot analysis, while Wnt3a-Myc, Wnt5a-Myc, and Wnt11-Myc (other major Wnt ligands in the *Xenopus* gastrula) were not (Supplementary Fig. 2b), but we could not detect endogenous Wnt8 by western blotting, maybe due to low sensitivity of the antibody to denatured Wnt8.

Fig1h requires quantification.

Ans: According to the reviewer’s suggestion, we quantified stained signals for Wnt8 and the two types of HS along the cell boundary, and calculated correlation coefficients between their distributions (new Fig. 1f-k). This quantification showed a statistically significant difference of correlation coefficients between Wnt8 with N-sulfo HS and Wnt8 with N-acetyl HS, as expected (p. 8, lines 139-143).

Figure 2

a,b quantification is needed.

Ans: According to the reviewer's suggestion, we quantified average intensities of stained signals for HS at the cell boundaries between two tracer-positive or tracer-negative cells, and statistically analysed the results (new Fig. 2d,e; see new Supplementary Fig. 5b for the method). This statistical analysis showed significant differences of N-acetyl or N-sulfo HS between *ndst1* mRNA or MO-injected and control cells, as expected.

d in general the differences are difficult to see as the basic expression levels differ. Furthermore, overexpression of *ndst1* mRNA is missing.

Ans: We responded the reviewer's suggestion by quantifying average signal intensities. Our results showed statistically significant differences between them, as expected (new Fig. 5a,c). In addition, we also examined the effect of overexpression of *ndst1* on the amount of endogenous Wnt8 (new Fig. 5b).

e, staining is not convincing. Non-*ndst1*-injected cells show also a strong beta-catenin localisation in the nucleus. Therefore, a quantification is missing.

Ans: We carried out *ndst1* mRNA and MO injection experiments, then quantified the ratio of nuclear localization of beta-catenin in *ndst1* mRNA or MO injected versus that in non-injected cell with *wnt8* overexpression (as illustrated in new Fig. 5d). The data showed that *ndst1* mRNA injection enhanced nuclear accumulation of beta-catenin at the cells far from the source of Wnt8; that is, low Wnt8 region (new Fig. 5e,f). On the other hand, *ndst1* MO injection reduced nuclear accumulation of beta-catenin at the cells close to the source of Wnt8; that is, high Wnt8 region (new Fig. 5g,h). These differences were statistically significant (Fig. 5f,h).

Could this experiment be repeated with Wnt8 MO?

Ans: The reviewer may have requested us to carry out an experiment to see whether *wnt8* MO reduces Wnt signalling assayed with nuclear localization of beta-catenin. However, this experiment seems to be very difficult to do, because when *wnt8* is knocked down in some part of *wnt8*-expressing cells, we hardly know how Wnt8 secreted from other cells is distributed in the embryo (please note that the nuclear beta-catenin analysis requires

side-by-side comparisons similar to Fig. 5e,g within the same embryo). Hence, it is almost impossible to look for a suitable region to observe.

If the reviewer intended *ndst1* MO, not *wnt8* MO, we performed the experiment as described above (Fig. 5g,h).

Is there a possibility to proof that sulphated HS is required for signalling in a more direct way?

Ans: To obtain more direct evidence that N-sulfo-rich HS is required for Wnt signalling, we performed three experiments as follows. The first one is improvement of nuclear beta-catenin staining. As already mentioned above, we succeeded to obtain clear results showing *ndst1* MO reduced nuclear b-catenin staining (Fig. 5g,h). The second one is TOP-Flash Wnt reporter experiments in the embryos. The data showed that overexpression or knockdown of *ndst1* enhanced or reduced, respectively, TOP-Flash luciferase activity, as expected (Fig. 5i-j). The third one is immunostaining of phosphorylated LRP6, which is a good indicator for signalosome formation. As already mentioned above, immunostaining of pLRP6 was reduced by *ndst1* MO (Fig. 7h). We believe that these results strongly suggest that N-sulfo HS is critical for activation of Wnt signalling.

Supplementary figure 2

b, this is a convincing staining. The author should discuss the similarities to the Huang & Niehrs, 2014.

Ans: We cited Huang and Niehrs (2014) and discussed the similarity of the Wnt8 distribution in the context of apicobasal polarity (p. 30, lines 525-527).

Supplementary figure 3

f, the authors use STED microscopy to measure the cluster size and find that the clusters do not differ. I find this experiment less interesting. However, one could use this microscopy technique to elucidate the colocalisation of HS with Wnt8.

Ans: As discussed for a similar comment by the reviewer #1, we agreed that the difference of the cluster size may be less informative because the size of IgM is comparable to the difference of the size between the two types of HS. Thus, we removed statistical analysis of the cluster sizes from the revised paper.

On the other hand, we suppose that it is difficult to use STED for evaluating colocalization because a strong laser beam used for emission depletion in STED system causes severe photobleaching and thus it is not suitable to detect relatively weak signals of endogenous Wnt8.

Figure 4

The authors observe internalisation of Wnt8 within seconds. this is very similar to studies in Hagemann 2014. In which compartments is Wnt8 internalised? Are these early endosomes, late endosomes, recycling endosomes?

Are these caveolae positive of Clathrin positive vesicles? Would this fit with the Kikuchi model?

Ans: We have cited Hageman et al., (2014) in the revised text, and discussed the similarity of the dynamic behaviour of Wnt8 internalization (p. 19, lines 339-341). As the reviewer asked, it is very interesting to know what kind of endosomes are involved in the internalization of Wnt8 as well as N-sulfo HS and glypicans. However, we have not yet found good antibodies for visualizing those endosomes in *Xenopus* embryos. Thus, we focused on caveolin and N-acetyl-rich and N-sulfo-rich HS in HeLa cells. The data showed that N-sulfo-rich HS but not N-acetyl-rich HS overlaps with caveolin (Supplementary. Fig. 13h,i). Considering the roles of N-sulfo-rich HS clusters in Wnt signalling as well as the internalization of HS, this data probably fits to the Kikuchi's model.

d, the authors change from Wnt8 to Fzd8 here. Therefore, the authors should provide evidence that Fzd8 is internalized together with Wnt8. Furthermore, it is unclear if Fzd8 co-localise similarly with sulphated HS but not with acetylated HS.

Ans: According to the suggestion by this reviewer, we examined internalization of Fzd8 associating with the exogenously expressed Wnt8-Myc (Supplementary. Fig. 12h,i). In this system, where we can easily distinguish the Wnt-source cells (“*” in Supplementary. Fig. 12h) and the receiving cells, we can judge Wnt internalization by detection of Wnt8-Myc puncta inside the receiving cells. Because we frequently observed that the endogenous Fzd8 was colocalized with internalized puncta of Wnt8-Myc, we concluded that Wnt8 and Fzd8 were actually co-internalized.

Regarding the relationship between Fzd8 and the two types of HS, we showed that the distribution of Fzd8 on the cell boundary was highly correlated with that of N-sulfo HS

but not of N-acetyl HS (Supplementary Fig. 12a-f). In addition, we also showed that knockdown of *ndst1* reduced Fzd8 on cell boundaries (Fig. 7g). These data suggest that the formation of signalosomes and the internalization of ligand-receptor complex occurred together with N-sulfo-rich HS.

Do the authors observe continuous endocytosis of Fzd or is this Wnt triggered?

Ans: Although we are also interested in this question, we consider that it is very difficult to answer this question due to technical limitations, such as an inducible expression system, in *Xenopus* embryos. However, we found that the signal of Fzd8 becomes high when Wnt8 is overexpressed (Supplementary Fig. 12g), suggesting that Fzd8 is stabilized or induced by Wnt8. This raises a possibility that Wnt8 and Fzd8 form a positive feedback loop, possibly leading to continuous endocytosis of Fzd as well as other signalosome components.

Figure5 + Suppl figure 10

The co-localisation data are presented in an excellent way – unfortunately partially hidden in the sup fig 10. The authors should put these findings in the centre of the paper. Co-localisation data should be backed-up with biochemical methods such as co-ip experiments. Additional, experiments are required to elucidate the molecular nature of the signalling / non-signalling platforms, a link to endocytosis is required and the consequences on signalling has to be shown more in detail

Ans: In accordance with the reviewer's suggestion, we have changed the place of the figures: previous Supplementary Fig. 10 to new Fig. 8. In addition, we performed similar quantification for endogenous Wnt8 (Fig. 1f-k), pLRP6 (Fig. 7a-f), and endogenous Fzd8 (Supplementary Fig. 12a-f).

Regarding with the biochemical analysis supporting the colocalization, we have tried to perform co-ip experiment using anti-N-acetyl or N-sulfo antibodies. Unfortunately, we could not obtain any specific band by this approach, probably because these antibodies are IgM, which is known to be hard to use for immunoprecipitation. Regarding with the detailed analysis on linking between two types of HS clusters and signalling, we found that N-sulfo HS is highly colocalized with caveolin (Supplementary Fig. 13h,i), which is suggested to be involved in canonical Wnt signaling (Yamamoto et al., 2006). In addition, a series of our new results using antibody for phosphorylated LRP6 clearly showed that the phosphorylation of LRP6, which is one of the most reliable markers of

the Wnt signalosome, was dependent on N-sulfonation of HS (Fig. 7h). These results provide us with a clearer model about the signalling on the cell membrane around the N-sulfo-rich HS cluster (Fig. 9c,d).

Reviewers' Comments:

Reviewer #1:

Remarks to the Author:

This revised manuscript concerns Wnt8 in *Xenopus* Development, and in particular the roles of heparan sulfate. The authors have included a significant amount of new data and revised some interpretations. In all this is an interesting set of data that should stimulate further work and be of great interest to the fields of developmental biology and proteoglycan metabolism.

There are just a few points that need to be raised.

1. Minor point. If an x-z scan of the data shown in Fig 1 is available it would help confirm that the orange arrowed material is internalised.
2. The Frzb data are among the most important in the manuscript, since the authors conclude that it preferentially binds to N-acetyl type structures in heparan rather than sulfated forms. This has consequences for Wnt signaling. Suppl Fig 14 is very helpful and important and some parts of it probably warrant inclusion in the main text. However, there is some concern that in Fig 8a, Frzb seems to co-localize with both NA and NS type heparan sulfate structures. It is surprising that the authors do not include glypican-4 MO here, since it should deplete the NA forms of heparan but not the NS forms. Frzb should then also be depleted and this would help confirm the underlying hypothesis of the authors.
3. It is recognized that the authors include supplement Fig 13 in response to another reviewer, but this data is concerning. I am not convinced of the codistribution between Wnt3a and N-sulfo HS rather than N-acetyl HS, or that between N-sulfo HS and either pLRP6 or caveolin. Do HeLa have separable N-sulfo and N-acetyl forms of HS? This is a highly transformed cell line where HS metabolism may be very atypical. It is hard to see how in panel "g" a Manders coefficient of approx. 0.6 can be derived from images such as those in "f" where the amount of codistribution is absolutely minimal. Similar concerns relate to panels "h" and "i". This data detracts from the overall message.

Finally, on a very positive note, much work has gone into this manuscript and it presents some novel ideas. It is striking that one interpretation of the data is that glypican core proteins somehow control the type of HS that is synthesized on them. This is for now not understood, but is a fascinating concept for the future.

Reviewer #2:

Remarks to the Author:

The revised paper reports on the identification of two types of clustered HSPGs (heparan sulfate proteoglycans), one of which serves as a site for Wnt signalosome assembly. The authors made a significant effort to respond to initial concerns and amended the manuscript. They were able to biochemically separate the N-sulfo (NS) and N-acetyl (NA)-rich clusters, using cellulose acetate membrane electrophoresis combined with alcian immunostaining. They confirmed that the observed Wnt8/NS-rich puncta correspond to Wnt signalosomes by immunostaining of a known Wnt co-receptor (anti-pLRP6). They also present Wnt reporter activity and pLRP antibody staining in addition to β -catenin nuclear localization assays in embryos with modulated *Ndst1* enzyme levels. In vivo data with *Xenopus* embryos were substantiated with the analysis of HeLa cells, in which colocalization of NS HSPG puncta with caveolin was demonstrated. The present data are of higher quality and accompanied by quantification. Overall, the paper has been substantially improved and should be published.

Minor comments:

--Lack of immunoblot analysis is discouraging, however, depletion experiments argue for antibody specificity. Lack of Wnt8 detection by immunoblotting likely reflects low abundance of endogenous Wnt8 rather than the poor reactivity of the antibody to denatured Wnt8 protein.

--lines 365-370--The authors should comment on the finding that Wnt8 increases Fzd8 protein levels ventrolaterally, contrary to the reports that Fzd8 transcripts are enriched dorsally.

--lines 197-200. The negative data from immunoprecipitation experiments should be removed, as they do not add anything of substance to the paper.

--The observed Wnt8 puncta in neighboring cells may be derived from a low amount of RNA diffused to these cells, rather than from the internalization of signalosomes. Can the authors exclude this alternative interpretation?

Reviewer #3:

Remarks to the Author:

The manuscript has been substantially revised and improved, and may now be accepted for publication in Nature Communications.

I am glad to see that several points have been solved. However, in the following there are some suggestions to be addressed:

The co-localization of pLrp6 with N-acetyl-HS / N-sulfo HS is satisfyingly performed. The staining shows a preference towards N-sulfo HS (Fig. 7a-f). The quantifications are useful.

The co-localization of caveolin with N-acetyl-HS / N-sulfo HS is not satisfying, however, the quantification shows a tendency towards N-sulfo HS/ Caveolin (Fig 13e-i). Could the staining be improved?

Fig. 7h : Immunostaining of pLRP6 in frog tissue was reduced by ndst1 MO knock-down. A quantification should be added.

The staining experiments for pLrp6 using HeLa cells in Fig 13f,g and the co-localisation study is poorly executed. I would suggest to remove them as they do not add anything to the argument.

Response to reviewers' comments

We thank all the reviewers for their helpful and constructive comments to improve our manuscript.

Reviewers' comments were pasted in blue and our response is written in black.

Reviewer #1 (Remarks to the Author):

This revised manuscript concerns Wnt8 in *Xenopus* Development, and in particular the roles of heparan sulfate. The authors have included a significant amount of new data and revised some interpretations. In all this is an interesting set of data that should stimulate further work and be of great interest to the fields of developmental biology and proteoglycan metabolism.

There are just a few points that need to be raised.

1. Minor point. If an x-z scan of the data shown in Fig 1 is available it would help confirm that the orange arrowed material is internalised.

We appreciate this comment. To answer this suggestion, we added an x-z plane image of the endogenous Wnt8 staining (new Supplementary Fig. 2d).

2. The Frzb data are among the most important in the manuscript, since the authors conclude that it preferentially binds to N-acetyl type structures in heparan rather than sulfated forms. This has consequences for Wnt signaling. Suppl Fig 14 is very helpful and important and some parts of it probably warrant inclusion in the main text.

We agree with this comment. However, the key experiment using Ndst1 on the Frzb's specificity has already been placed in main figure (Fig. 10a). In addition, there is a limitation of the number of main Figures. Therefore, we would like to keep Supplementary Fig 14, as it was.

However, there is some concern that in Fig 8a, Frzb seems to co-localize with both NA and NS type heparan sulfate structures. It is surprising that the authors do not include glypican-4 MO here, since it should deplete the NA forms of heparan but not the NS forms. Frzb should then also be depleted and this would help confirm the underlying hypothesis of the authors.

As this reviewer pointed out, some Frzb signals appear to co-localize with NS puncta in the photo shown in Fig. 8a. However, as described in Fig. 8a (new Fig. 9a), the quantification shows that Frzb preferentially colocalized with N-acetyl-rich HS, compared to N-sulfo-rich HS. Thus, we conclude

that Frzb preferentially co-localize with NA. Since our biochemical analysis suggests that N-sulfo-rich HS clusters still contain some content of N-acetyl HS, it seems plausible that some Frzb associating with N-acetyl HS remained in NS puncta could be observed in new Fig. 9a.

In addition, as the reviewer suggested, we also think that knockdown of glypican-4 is an interesting experiment. However, glypican-4 is a common core protein for both N-acetyl-rich and N-sulfo-rich HS, and thus its interpretation might be vague. In conclusion, we believe that the specificity of Frzb to N-acetyl-rich HS has sufficiently been supported by overexpression of Ndst1 (new Fig. 10a and Supplementary Fig. 13c).

3. It is recognized that the authors include supplement Fig 13 in response to another reviewer, but this data is concerning. I am not convinced of the codistribution between Wnt3a and N-sulfo HS rather than N-acetyl HS, or that between N-sulfo HS and either pLRP6 or caveolin. Do HeLa have separable N-sulfo and N-acetyl forms of HS? This is a highly transformed cell line where HS metabolism may be very atypical. It is hard to see how in panel "g" a Manders coefficient of approx. 0.6 can be derived from images such as those in "f" where the amount of codistribution is absolutely minimal. Similar concerns relate to panels "h" and "i". This data detracts from the overall message.

We appreciate this suggestion. Considering together with similar concerns raised by another reviewer, we decided to omit pLRP6 data in HeLa cells (Supplementary Fig. 13e-g). However, we would like to keep panels "h" and "i", because these data were shown to answer one of the main concerns by the reviewer 3. Because similar experiment is difficult to perform with *Xenopus* embryo due to cross-reactivity of the antibody, we decide to keep this figure in this re-revised version. Regarding HS in HeLa cells, we have already described distinct distributions of NAH46 and HepSS-1 (page 10, lines 309-312, new Supplementary Fig 3f, data itself was not changed).

Finally, on a very positive note, much work has gone into this manuscript and it presents some novel ideas. It is striking that one interpretation of the data is that glypican core proteins somehow control the type of HS that is synthesized on them. This is for now not understood, but is a fascinating concept for the future.

As this reviewer pointed out, the mechanisms by which the type/modification of HS is determined should be addressed in the future works. In accordance with this suggestion, we added the following sentence in Discussion: "Therefore, glypican core proteins appear to control the type of HS that is attached to them." (page 28, lines 841-842).

--

Reviewer #2 (Remarks to the Author):

The revised paper reports on the identification of two types of clustered HSPGs (heparan sulfate proteoglycans), one of which serves as a site for Wnt signalosome assembly. The authors made a significant effort to respond to initial concerns and amended the manuscript. They were able to biochemically separate the N-sulfo (NS) and N-acetyl (NA)-rich clusters, using cellulose acetate membrane electrophoresis combined with alcian immunostaining. They confirmed that the observed Wnt8/NS-rich puncta correspond to Wnt signalosomes by immunostaining of a known Wnt co-receptor (anti-pLRP6). They also present Wnt reporter activity and pLRP antibody staining in addition to β -catenin nuclear localization assays in embryos with modulated Ndst1 enzyme levels. In vivo data with *Xenopus* embryos were substantiated with the analysis of HeLa cells, in which colocalization of NS HSPG puncta with caveolin was demonstrated. The present data are of higher quality and accompanied by quantification. Overall, the paper has been substantially improved and should be published.

Minor comments:

--Lack of immunoblot analysis is discouraging, however, depletion experiments argue for antibody specificity. Lack of Wnt8 detection by immunoblotting likely reflects low abundance of endogenous Wnt8 rather than the poor reactivity of the antibody to denatured Wnt8 protein.

We are happy to know that this reviewer admitted the specificity of the antibody. We agree with the possibility raised by this reviewer.

--lines 365-370--The authors should comment on the finding that Wnt8 increases Fzd8 protein levels ventrolaterally, contrary to the reports that Fzd8 transcripts are enriched dorsally.

Thank you for the comment. To answer this comment, we added following statement in the text: "Therefore, Wnt8 might increase Fzd8 protein levels in the ventrolateral region, even though *fzd8* transcripts are enriched dorsally⁵⁰. In fact, Fzd8 staining was increased in the mDMZ by overexpression of Wnt8 (Supplementary Fig. 11g). Considering the dorsally enriched *fzd8* transcripts⁵⁰, Wnt8 possibly stabilizes Fzd8 protein rather than enhances the expression of *fzd8*." (pages 20-21, lines 599-622)

--lines 197-200. The negative data from immunoprecipitation experiments should be removed, as they do not add anything of substance to the paper.

According to this suggestion, we deleted the description about the negative data of immunoprecipitation. Thank you.

--The observed Wnt8 puncta in neighboring cells may be derived from a low amount of RNA diffused to these cells, rather than from the internalization of signalosomes. Can the authors exclude this alternative interpretation?

As this reviewer pointed out, diffusion of a low amount of mRNA may cause Wnt8 puncta in neighboring cells. However, we believe that our experiments were not the case by several reasons. First, if mRNA injected into a blastomere leaks into another blastomere, mRNA should be diffused in a gradient manner. Thus, we can exclude a case where mRNA is diffused from injected blastomere, by judging whether protein expression is in a graded manner. However, in the case of the experiment pointed by this reviewer, we observed clear difference in Wnt expression level between Wnt-overexpressing cells and neighboring cells. Second, we have already confirmed that mRNA carefully injected into the animal pole region after late 4-cell stage never shows leakage into another blastomere (Mii & Taira, 2009).

--

Reviewer #3 (Remarks to the Author):

The manuscript has been substantially revised and improved, and may now be accepted for publication in Nature Communications.

I am glad to see that several points have been solved. However, in the following there are some suggestions to be addressed:

The co-localization of pLrp6 with N-acetyl-HS / N-sulfo HS is satisfyingly performed. The staining shows a preference towards N-sulfo HS (Fig. 7a-f). The quantifications are useful.

We are glad to hear a positive comment for it.

The co-localization of caveolin with N-acetyl-HS / N-sulfo HS is not satisfying, however, the quantification shows a tendency towards N-sulfo HS/ Caveolin (Fig 13e-i). Could the staining be improved?

We are sorry for this point. Although we repeated the experiment several times to examine the colocalization of caveolin and HS, the staining of caveolin was not clearly improved. Thus, we consider that it is quite difficult to improve the quality of this staining. In addition, since this antibody did not work for *Xenopus* caveolin, we cannot perform similar experiment in *Xenopus* embryo, which may show a clearer result. Considering that the quantification shows a tendency towards N-sulfo HS/ Caveolin, as this reviewer admitted, we would like to keep these data without further improvement.

Fig. 7h: Immunostaining of pLRP6 in frog tissue was reduced by *ndst1* MO knock-down. A quantification should be added.

We appreciate this comment. We performed a quantification with a statistical test for the pLRP6 puncta, which was resulted as expected (new Fig. 8i).

The staining experiments for pLrp6 using HeLa cells in Fig 13f,g and the co-localisation study is poorly executed. I would suggest to remove them as they do not add anything to the argument.

We appreciate this suggestion. Together with similar concern raised by another reviewer, we decided to omit pLRP6 data in HeLa cells (Supplementary Fig. 13e-g).